# Characterizing Mechanisms for Factual Recall in Language Models

**Qinan Yu**        **Jack Merullo**        **Ellie Pavlick**
Brown University
Department of Computer Science
{qinan_yu,jack_merullo,ellie_pavlick}@brown.edu

## Abstract

Language Models (LMs) often must integrate facts they memorized in pretraining with new information that appears in a given context. These two sources can disagree, causing competition within the model, and it is unclear how an LM will resolve the conflict. On a dataset that queries for knowledge of world capitals, we investigate both distributional and mechanistic determinants of LM behavior in such situations. Specifically, we measure the proportion of the time an LM will use a counterfactual prefix (e.g., "The capital of Poland is London") to overwrite what it learned in pretraining ("Warsaw"). On Pythia and GPT2, the training frequency of both the query country ("Poland") and the in-context city ("London") highly affect the models' likelihood of using the counterfactual. We then use head attribution to identify individual attention heads that either promote the memorized answer or the in-context answer in the logits. By scaling up or down the value vector of these heads, we can control the likelihood of using the in-context answer on new data. This method can increase the rate of generating the in-context answer to 88% of the time simply by scaling a single head at runtime. Our work contributes to a body of evidence showing that we can often localize model behaviors to specific components and provides a proof of concept for how future methods might control model behavior dynamically at runtime.

## 1  Introduction

Large Transformer Language Models (Vaswani et al., 2017) (LMs) store information from pretraining which they can recall at inference time to generate text. This is paired with the exceptional ability of models to use provided context in order to produce coherent text that incorporates new facts. However, facts that are memorized in pretraining and facts that are provided in-context can often compete with each other; in some cases it might

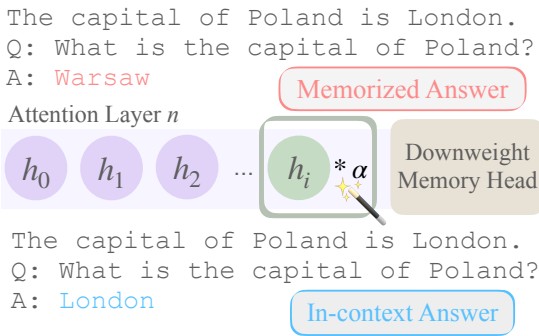

Figure 1: We find that individual attention heads can play specific roles in using context information vs. recalling facts. By up or downweighting these heads, we can often control whether LMs use information from context which conflicts with its pretraining knowledge. For example, downweighting the memory attention head causes the model to prefer "London" above.

be desirable that the model ignores facts from pretraining (e.g., updating outdated information with a prompt), while in others we want the model to prefer what it learned in pretraining (e.g., ignoring false information in prompt injections). Currently, little is understood about the factors and mechanisms that control whether an LM will generate text respecting either the in context or memorized information.

Recent work in mechanistic interpretability aims to deeply explain the internal processes in LMs, allowing us to interpret the contributions of individual model components to the final predicted text (Olah et al., 2020; Wang et al., 2022; Nanda, 2022). We can use tools from these studies to shed light on the model components that are responsible for pushing the model more towards either memorized or contextual information. We study this relationship using a task that requires predicting a capital city in the face of conflicting information (see Figure 1). We measure how often a model will answer that the capital city of a country is the *in-context* counterfactual (e.g., London) vs. the *memorized* (ground

truth) city (Warsaw) it learned in pretraining. Our study consists of two key sets of experiments.

First, in Section 5, we investigate how distributional features of the pretraining data influence behavior. We find that the frequency of a fact in the pretraining corpus strongly correlates with model behavior. This analysis reveals several key findings: (1) The more frequently a country appears in pretraining, the more likely the LM is to generate the memorized capital city; (2) The more frequently the in-context city (i.e., the one with which we want to overwrite) appears, the less likely the model is to use it, regardless of the frequency of the country; (3) Larger models (up to the scale tested: 2.8b parameters) are less likely overall to use in-context information and prefer the memorized answer, even when the fact is less frequent.

Next, in Section 6, we use head attribution (Elhage et al., 2021; Nostalgebraist, 2020; Nanda, 2022) to show that we can localize promotion of the memorized or in-context answer to individual attention heads. By either upweighting or downweighting the head by a scalar value, we can control which answer the model prefers. In the most successful case, downweighting the memory head allows us to increase the rate of the in-context city to 88% while reducing the amount of memorized predictions to 4% on the world capitals task. In a qualitative analysis of the weights of this head (Section 6.4), we show that it specifically promotes geographic information. We find that forcing the opposite behavior, i.e., promoting the memorized answer, is more difficult, and that the mechanism these heads use doesn't necessarily generalize well (§6.5). Still, the method we discover is surgical, and only requires scaling a single head (0.00001% of Pythia-1.4b's parameters), suggesting that components within an LM may specialize for specific predictable functions, and providing a promising avenue for understanding the internal workings of LMs and further techniques for model editing.

## 2   Related Work

The impressive, but sometimes unpredictable successes of LMs on performing tasks described in context (Brown et al., 2020) has spurred intense interest in the factors that allow models to solve tasks this way. Studies on pretraining datasets have found that higher pretraining term frequency is positively correlated with task performance on factual association tasks (Kandpal et al., 2022) and numer-

ical reasoning (Razeghi et al., 2022), and relates to work on memorization vs. generalization in LMs (Hupkes et al., 2022). Haviv et al. (2023) analyze mechanisms used to recall memorized information by studying idiom generation. Model size is also shown to be a factor that affects tendency to use memorized vs. in context information (Wei et al., 2023). Previous work has also examined how deeply LMs interact with context during in-context learning (Min et al., 2022; Xie et al., 2022). Other work has focused on LMs' abilities to consider counterfactual or hypothetical contexts (Li et al., 2023; Qin et al., 2019), with mixed results in overwriting pretraining memory.

Our work is built heavily on previous work in mechanistic interpretability, which aims to reverse engineer model computations into human understandable components (Nanda et al., 2023; Elhage et al., 2021; Wang et al., 2022). While knowledge from pretraining has been found to be stored in the feedforward (MLP) sublayers (Meng et al., 2023; Geva et al., 2021; Kobayashi et al., 2023; Dai et al., 2022), more recent work has also clarified the role of attention in this same process: Geva et al. (2023) find that attention heads extract facts from an earlier mentioned subject token (e.g., Poland) when required. This naturally sets up our study, which also considers attention heads as the source of the competing effect between copying the counterfactual from earlier in context vs. extracting the memorized fact from an earlier subject token. A core technique in these works is projecting activations from model components into the vocabulary space to make claims about their roles, which we generically refer to here as *logit attribution* (Nostalgebraist, 2020; Wang et al., 2022; Merullo et al., 2023; Belrose et al., 2023; Dar et al., 2022; Millidge and Black, 2022). We leverage this technique to localize attention heads which tend to promote either context or memorized information (§6).

## 3   Task Design

We study the mechanism that language models use when given counterfactual information in context. For our analysis, we focus on a simple zero-shot task that requires producing the capital city for a given country, which serves as a representative example of the type of common facts that a language model could learn in pretraining. It consists of 248 country capital pairs with 6 additional aliases and their respective capitals. To create counterfactuals

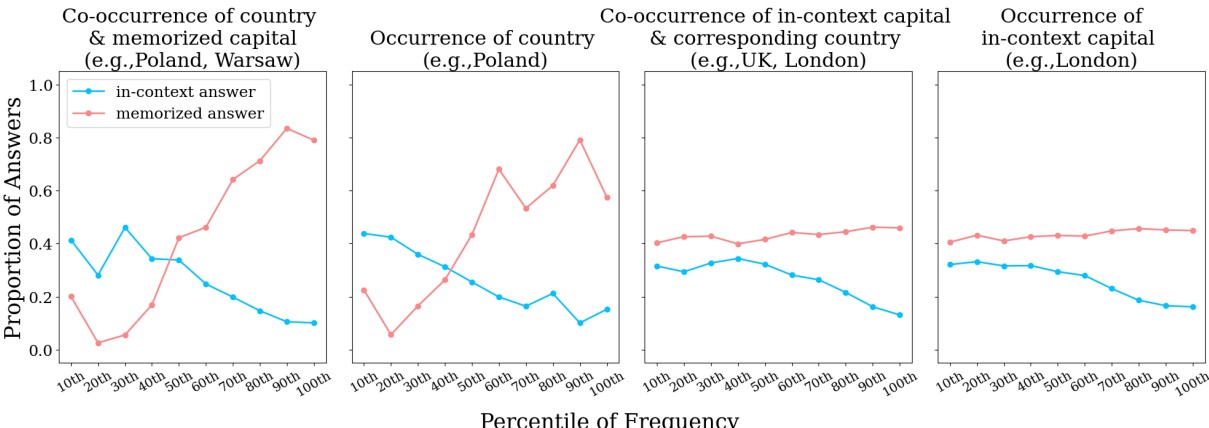

Figure 2: Results from total 62,992 inputs of every country paired with every counterfactual capital 1. We break down all the inputs into 10 percentile bins from the least to most frequent by four frequency criteria. Every percentile contains around a total of 6300 examples. The first two graphs reflect frequency based on the country (e.g., Poland). The upward trajectory of the red lines show the positive correlation between the proportion of memorized answer predictions and frequency. The last two graphs reflect the frequency of the *in-context capital* (e.g., London). The drop of the blue lines across all the four graphs show the negative correlation between proportion of in-context answer predictions and term frequency.

in the dataset, we pair up every country with the rest of the 247 capitals using the following format:

```
The   capital   of   {country}   is
{in-context city}.
Q: What is the capital of {country}?
A:
```

For example, we can fill in **{country}** with Poland and **{in-context city}** with London. The model has learned that the capital is Warsaw from pretraining, but with the in-context prompt, the task becomes ambiguous between whether it should output the known answer Warsaw or overwrite it with London. We query the model to generate a full sentence to determine which of the above task interpretations it preferred. We define Poland as the *country* and London as the in-context answer. Correspondingly, we define Warsaw, the capital of Poland, as the memorized answer. If London is included in the sentence, then we consider the model to have produced the in-context answer. If the model generates Warsaw, then it is considered as the memorized answer. In total, we have 62,992 such pairs of country and capital.

The World Capital dataset is able to provide a clean analysis giving a unique memorized and in-context answer. Language models perform well on this task producing one of the two expected cities at least 80% of the time (varying depending on model size). The lack of noise in the responses makes the

task a good choice for cleanly diagnosing model preference for memorized vs. in-context answers.

## 4 Models

We analyze the overwriting behavior primarily on the Pythia (Biderman et al., 2023) models as well as GPT2 series (Radford et al., 2019). The Pythia models are trained on the Pile (Gao et al., 2020) which we have full access to, allowing us to relate model behavior to frequency effects in the data. While we don't have access to the pretraining data of GPT2, we still report results, using the Pile frequencies as an approximation of what GPT2 might have seen.

## 5 Effect of Term Frequency on Model Tendency to Use Memorized Facts

### 5.1 Experimental Setup

We hypothesize that the model will be less likely to overwrite information about more frequently appearing country/capital names. The number of in-context predictions will increase when the term frequency descreases. The number of memorized predictions will increase when the term frequency increases. To test this hypothesis, we search through the the pretraining corpus of the Pythia model (i.e., the Pile (Gao et al., 2020), which contains 210 million text documents) in order to compute the term frequency of country and city names.

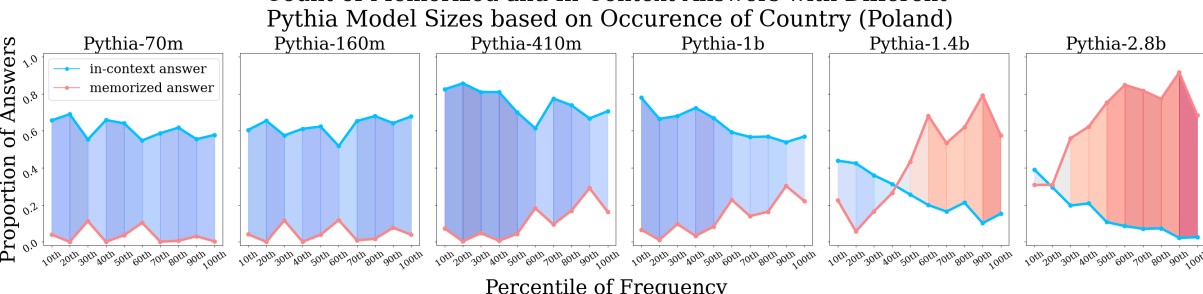

Figure 3: The proportion of in-context and memorized answers decomposed by the frequency of *country*(Poland) across all Pythia models of different sizes (the $2^{nd}$ graph in figure 2). The upward trend of the red lines shows as the model size increases, the model predicts more memorized answers. Blue and red shading indicates that the amount of in-context or memorized answers is higher, respectively. We find that as models get bigger, they first memorize more frequent capitals before the lower frequency ones.

We search both for the frequencies of the individual country and city names, as well as their co-occurrences in the dataset. Co-occurrence is measured by whether a country and a city appear together in the same document. We split the occurrences and co-occurrences into 10 percentile bins, with the 0th bin containing the least frequent 10%, and the 9th bin containing the top 10% most frequent terms. Every bin includes around 25 countries and capitals. We mix every country with all the other 248 capitals to form prompts. We have in total around 6200 instances per bin (given 63k prompts). To give some qualitative examples, capitals like `Beijing` are in the top percentile bin as measured by occurrence, while capitals like `Akrotiri` and `Dhekelia` are in the bottom. For the co-occurrences between country and capital, ⟨`China`, `Beijing`⟩ is in the top percentile and ⟨`Guinea-bissau`, `Bissau`⟩ is in the bottom percentile.

We run the counterfactual world capital data through both the Pythia models as well as the GPT2 series of models. We generate the a full sentence by decoding the output. We count the number of times the in-context and memorized answers appear in the decoded sentences and plot these counts as a function of the percentile bins described above.

### 5.2 Results

As the frequency for the *country* increases, there is more knowledge stored about the country during pertaining. Therefore, we intuitively expect to see that models are more inclined to predict the memorized answers as the frequency goes up. Figure 2 supports this intuition. We can see a clear upward trend in the pink line, reflecting the increasing pro-

portion of the memorized answers as a function of the increase in term frequency. When the *country* is more prevalent in the training data, the model has a greater tendency to predict memorized answers.

We also observe a relationship between the frequency of the in-context capital and the model's predictions. As the frequency for either the country or the in-context capital increases, the number of in-context answer predictions decreases. This is demonstrated by the drop of the blue lines in Figure 2. When the given in-context capital is more prevalent in the training data, for example `Beijing`, the model tends to predict the memorized answer. However, when the given in-context capital is less prevalent, such as `Palikir`, the model is more likely to predict the in-context answer. We ran the same experiments across all the Pythia and GPT2 models of different sizes (see Appendix A) and see the same frequency effect, especially in larger models.

Figure 3 shows the increase in sensitivity to frequency with respect to model size. We find that as models increase in size, they become more likely overall to produce the memorized answers rather than in-context answers, and that this occurs with the most frequent countries. That is, as larger models become more likely to produce the memorized answer, the changes are not evenly distributed across frequency bins. Rather, a strong memorization bias is observed first for more frequent terms, and then as models get larger, this extends to increasingly lower frequency terms. This can be observed in transition from blue shading (more in-context answers) to red shading (more memorized answers). See Appendix B.1 for results showing this effect with respect to the frequency of cities

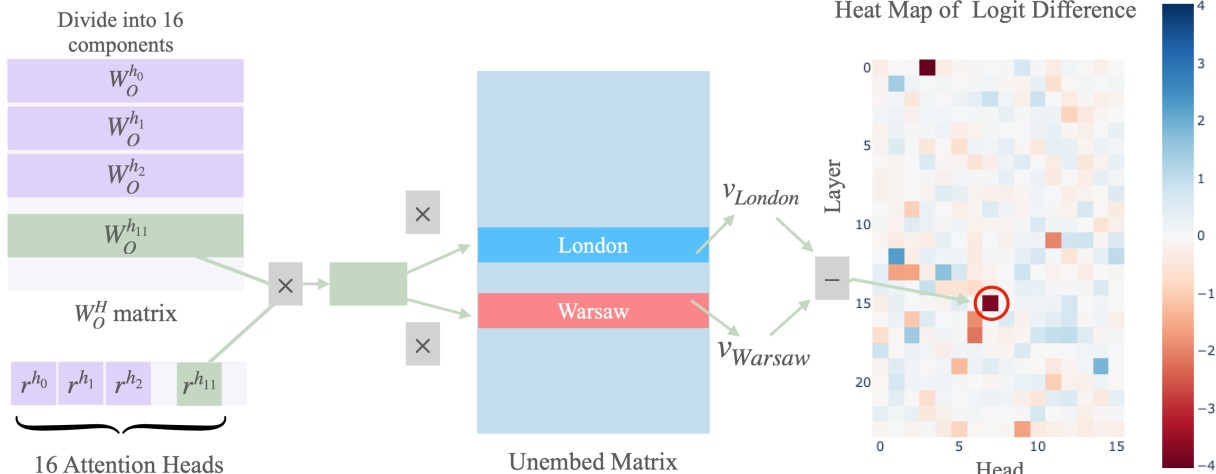

Figure 4: The head attribution method showing the logit difference calculation for layer 15, head 7 in Pythia-1.4b on the example from Figure 1. Pythia-1.4b has 24 layers and 16 heads for each layer, totaling 384 heads to check. We obtain the memory head and in-context head in the following way: We divide the output weight matrix from an attention layer ($W_O^H$) into 16 components (one for each head) (Elhage et al., 2021) Then, we take the dot product between each head $i$ of the and the $i^{th}$ component of the weight matrix. Afterward, we extract the corresponding vectors in the unembedding matrix for the memorized answer (e.g., Warsaw) and in-context answer (e.g., London). We dot product the projected head vector with the two vectors respectively, giving us a scalar value representing the logit for each of those words represented by the head. Subtracting these two scalars give us the logit difference of two answers from one specific head. Blue in the heatmap indicates that the head is promoting in-context answer and red indicates the head is promoting memorized answer.

and co-occurrences, where we observe the same trend.

## 6 Identifying and Manipulating Mechanisms for Recall

So far we have shown that (larger) models tend to have a preference to use the answer they have memorized. In this section we ask if there is a specific mechanism within the model that controls whether the memorized or in-context answer is generated, and whether that can be isolated from more general language generating abilities. Because the task boils down to whether the model copies information that was provided in context or not, we focus on analyzing the roles of specific attention heads. Prior work has demonstrated the importance of attention heads for performing copying tasks (Wang et al., 2022; Elhage et al., 2021) as well as recall from memory (Geva et al., 2023), which motivates our analysis of attention heads. We perform this analysis on only the largest models Pythia-1.4b, Pythia-2.8b, as well as GPT2-xl (see Appendix D).

### 6.1 Head Attribution

The idea behind logit attribution techniques (Nostalgebraist, 2020; Wang et al., 2022; Nanda, 2022) is to interpret activations or weights in a language

model in terms of the vocabulary space. These methods work by using the unembedding matrix (i.e., language modeling head) in order to understand the role of a given component for a given task. This is built on the premise that the final hidden state of the model is the summation of the outputs of all of the components before it (Elhage et al., 2021). That is, every layer of output can be traced back and decomposed as the contribution of each sublayer up to that point. We use head attribution to test whether individual heads tend to promote either the in-context capital or the memorized capital. Using this method, we are able to find a single head in each model that primarily controls the use of memorized information[1].

In Figure 4, we illustrate the method. The additive update made by the attention layer is composed of the individual updates of each attention head after it is passed through the $W_O^H$ output matrix within the attention layer. We can project the $i^{th}$ head into the space of the residual stream by multiplying with the $i^{th}$ ($d_{head}, d_{model}$) slice of this matrix (see Appendix C) and then multiplying with the unembedding matrix to get the logit values for the memorized and in-context city tokens. We subtract

---

[1]This is not to say that this is the *only* job of this head in general, or that these are the only heads that play this role.

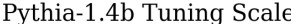

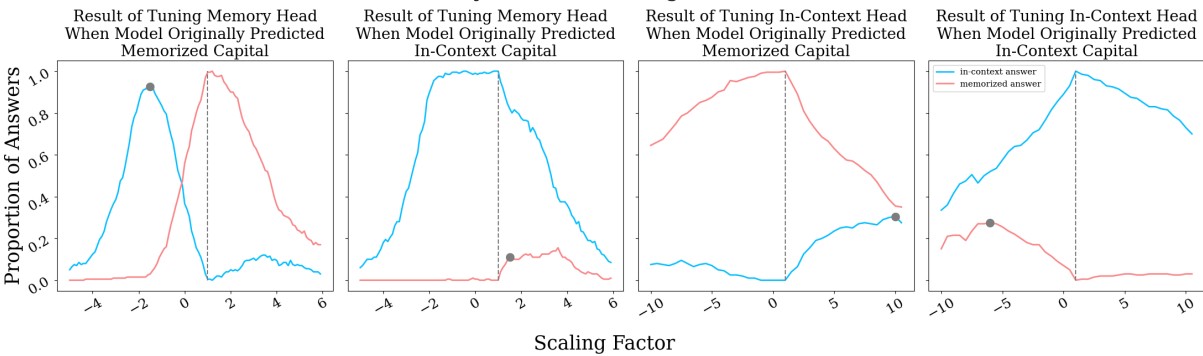

Figure 5: With the chosen memory head (15.7) and in-context head (19.14), we apply a multiplicative factor ($\alpha$) to measure the effect on producing either memorized or in-context answers. This was performed on two 100 example tuning sets (§6.1). The first graph demonstrates the most successful case of intervention. By tuning the memory head (15.7) value by $\alpha = -0.7$, can flip 86% of the examples from originally predicting memorized answers to predicting in-context answers. The dotted line shows no intervention ($\alpha = 1$). The gray dot shows the value of $\alpha$ that produces the best results according to our criteria.

these two scalar values to get the *logit difference* (see Wang et al. (2022)).

Intuitively, this logit difference captures the effect the head has in promoting one word (relative to another) to be output as the final prediction. This provides us a practical way to calculate the the role of each head, and find heads that consistently push the model towards the memorized or in-context answer.

**Data:** To identify specific heads, we randomly sample 10 examples from each percentile for which each model predicts the in-context answers and another 10 examples for which each model predicts the memorized answers. Thus, in total, we obtain 100 examples on which the original model predicts in-context cities and 100 examples on which it predicts memorized cities. We run these 200 examples through the model in batches of 5 and use head attribution to extract the logit difference between each head in every layer. We observe that there is a variation in the roles of every head throughout the batches, but we identify a series of heads that consistently push the model towards one answer or the other.

## 6.2 Effect of Tuning Individual Attention Heads

Using head attribution, we identify two different types of heads: **memory heads** and **in-context heads**. The memory heads promote the prediction towards the memorized answer and the in-context heads promote the predictions towards the in-context answer. These heads are shown on the

righthand side of Figure 4, which plots the relative effect of each head at each layer for promoting the in-context vs. memorized answers.

Since these heads heavily contribute to the logit increase for one of the two answers, we hypothesize that multiplying the value vectors by a scalar will enable us to increase or decrease the effect of each head. Let this multiplicative value be $\alpha$. We hypothesize that tuning up the memory head will increase the number of answers that contain the ground truth answer, while tuning it down will increase the number that contain the in-context answer. The opposite should hold for the in-context head.

With this assumption, we apply the scaling intervention on the series of potential memory heads and in-context heads on the 200 sampled examples. From the series of potential heads, we pick the head that has the strongest effect in the intended direction. See Appendix F for results using an alternative memory head. For example, for the in-context head, this effect is measured by the proportion of times the head changes the original memorized answers into in-context answers at it's optimal $\alpha$. The analogous process is used to find and tune the memory head. Therefore, we identify one memory head and one in-context head (see Figure 5, Appendix C.3), each with their optimal $\alpha$, as determined via tuning on the development set.

Figure 5 shows the effect of the $\alpha$ parameter on the proportion of in-context vs. memorized answers for both the memory and in-context heads on Pythia-1.4b. Tuning the memory down has a

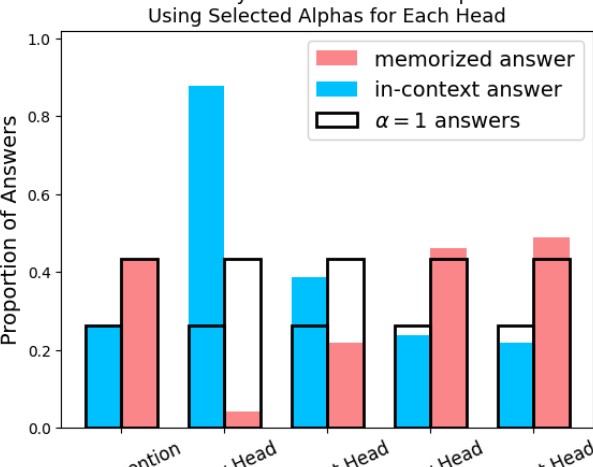

Figure 6: We apply the chosen memory head (15.7) and in-context head (19.14) and the chosen respective scale from Figure 5, we apply the scaling intervention on all of the 62,992 examples. Negatively tuning the memory head produced the most successful result.

strong effect on the generated text, flipping more than 80% of the predictions to the given in-context answers, and preventing the model from ever producing the memorized answer. The other interventions show positive but weaker results. In general, the in-context head is less effective at flipping predictions, and promoting the memorized answer is more difficult than promoting the in-context answer.

## 6.3 Results of Interventions on the World Capital Dataset

Figure 6 shows the intervention results on the full world capital dataset with selected memory and in-context heads and their respective $\alpha$. The result aligns with our expectations. Negatively tuning the memory head drastically increases proportion of the in-context answers. Specifically, whereas the model originally predicted in-context answers 26% of the time and memorized answers 43% of the time, after our intervention, the model predicts in-context answers 86.2% of the time and memorized answers only 4% of the time. Note that, on Pythia 1.4b, scaling a single head is analagous to modifying 0.00001% of model parameters. This suggests that this head plays a specific role in using the memorized answer in this task. Positively tuning the memory head also increases the memorized answer prediction to 50%. Positively tuning the in-context head pushes the model in the expected

direction but has a more muted effect: increasing the amount of in-context answers by 12% but dropping the amount of memorized answers by about 20%. We observe that changing in-context predictions to memorized predictions is more difficult. In the fourth and fifth column, when positively tuning the memory head and negatively tuning the in-context head, we hope to increase the proportion of memorized answers. While there is some increase, it is less profound, only increasing 6%. Given the connection between facts learned in pre-training and the MLP layers (Geva et al., 2021; Meng et al., 2023; Merullo et al., 2023), it's possible that tuning attention alone is not enough to see higher performance in this setting.

We break down the intervention results from Figure 7 into term frequency percentile bins as in Section 5. We focus on the occurrence count of the country and the occurrence count of the in-context capital (London). We select two interventions–negatively tuning memory head and positively tuning in-context head in Pythia-1.4b–both of which should increase the in-context answers and decrease the memorized answers. We find that the intervention on the memory head overcomes the previously-described frequency effects. Specifically, the dashed blue and pink lines are flat across percentiles. When positively tuning the in-context head, we observe that the frequency effects remain, and thus the intervention is not fully successful. In particular, even after intervention, the memorized answers are still positively correlated with term frequency, and the in-context answer is negatively correlated with frequency. Most prominently, tuning the in-context head does not substantially increase the number of in-context answers when the in-context city is high frequency (no mitigation of the city frequency effect) as shown by the blue lines in the 4th graph.

## 6.4 Head Analysis

Tuning the memory head down inhibits the models' abilities to promote the memorized capital city as we have shown in the previous section. In this section, we explore why this is the case by analyzing the memory head weights. We find the selected memory head (15.7) promotes geography related tokens in the output space, suggesting that this head is responsible for this information as opposed to a more abstract 'truthfulness' direction.

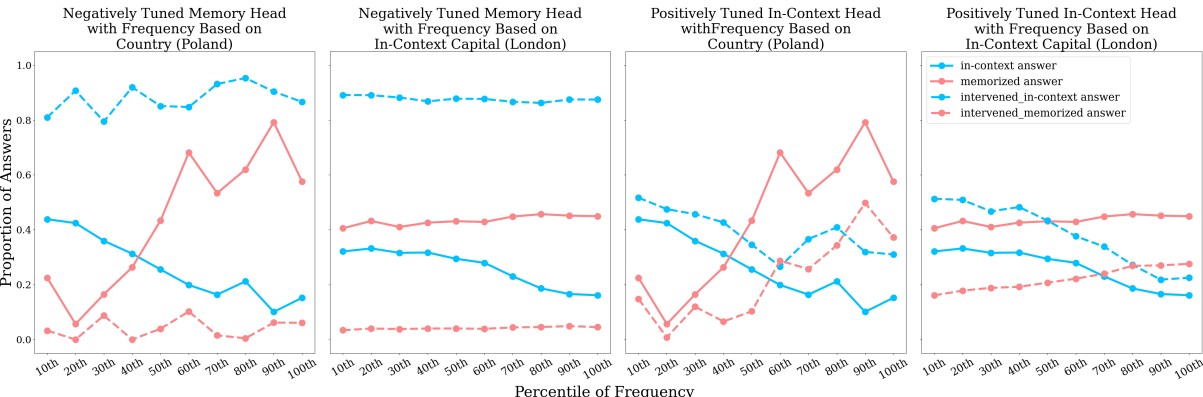

Figure 7: The frequency effect referred to in Section 5 disappears when we tune down the memory head, showing the success of this strategy. Positively tuning the in-context head shows decent success on lower frequency countries/capitals, but actually causes performance to fall apart in the higher frequency bins. The solid lines show the original predictions and the dotted lines show predictions after the intervention.

**Singular Vector Decomposition** The product of the Value and Output weight matrices in an attention head form the OV matrix (Elhage et al., 2021), which controls how attending to some token affects the residual stream. Following Millidge and Black (2022); Belrose et al. (2023), we can decompose the $OV$ matrix into $\mathbf{SVD(OV)} = \mathbf{USV_h}$ where $\mathbf{V_h}$ is the unitary matrix of the right singular vectors representing the subspaces the given head writes into the residual stream (as opposed to the Value weight matrix in $\mathbf{OV}$). See Millidge and Black (2022); Belrose et al. (2023) for more information.

The $i$th singular vector has the same size as the residual stream; if we decode this vector into the vocab space of the model with the unembedding matrix (the LM head) we can observe the semantic clusters that a given head most strongly promotes. Since the singular vectors are ordered, we know that the first singular vectors are the most important for the head. We qualitatively define the semantic clusters promoted by each head by looking at the top $k$ tokens decoded from each singular vector.

We compare the memory head 15.7 with the context head 19.4. If the memory head specifically promotes geographical information, we should see clear emphasis on this information that is not present in the context head. In Table 1, we decode the top 10 tokens from the first five singular vectors in each head and find that many of the memory head tokens are geographically focused. The trend becomes even more clear when comparing all singular vectors (see Appendix G). It should be noted that the alternate memory head studied

in Appendix F is not as interpretable in the vocab space, despite giving similar intervention results. Understanding the contribution of such heads is an interesting direction to future work.

### 6.5 Generalizing to New Data

We further explore the domain specificity of the memory (and context) head by applying the method to the COUNTERFACT dataset (Meng et al., 2023), which queries factual knowledge from multiple domains. We apply the same scale intervention to the heads in the discovered mechanism. We focus on the paraphrase task of the dataset in the zero-shot setting. For example,

```
Apple A5 is developed by Google.
Apple A5 is created by
```

The dataset replaces the memorized answer (`Google`) with a counterfactual (`Apple`). We filter the dataset down to examples where the model predicts the ground truth answer when the counterfactual prompts are not injected. We apply the same memory head, in-context head and their respective intervention scale on the counterfactual dataset. That is, we do no additional data-set specific analysis or tuning. We find that, despite the memory head's high impact on the world capital dataset (increasing the proportion of *in-context answers* by 60%) it doesn't generalize to the COUNTERFACT dataset. In both cases of interventions, both the proportion of memorized answers and in-context answers decrease. The model produces a higher proportion of invalid answers compared to the intervention on the world capital dataset. This

| Memory Head (15.7) |
| --- |
| ' LW', ' Wade', ' WI', 'liche', 'ienne', ' ell', 'owe', 'iale', 'uelle', 'ｅｔｅ' |
| ' Italian', 'Italian', ' Italy', ' Ital', ' Io', ' Giovanni', ' pasta', 'Io', ' Giul', ' Naples' |
| 'WA', 'WS', ' WA', 'owa', 'ws', 'wa', 'Ws', ' Wa', 'pora', ' WI' |
| ' WM', 'WM', 'wm', 'mw', 'w', 'nw', ' w', ' MW', ' Minnesota', 'WN' |
| ' Guatemala', ' Guatem', 'usta', 'osta', ' Tampa', ' Brazil', 'ativa', ' Bah', ' Tamil', 'Brazil' |

| In Context Head (19.4) |
| --- |
| '.', '.;', '.\u200b', '.*', '.:', '.-', '.),', '.?', '.);', '.).' |
| 'ilogy', 'vex', '必', 'xspace', 'verages', 'loat', '?', 'や', ' cres', 'HPP' |
| 'tron', '.%', '._', '————', ' Salem', ' Telesc', 'bsy', '".',", 'olean', 'inn' |
| 'ometown', 'LLY', 'suit', '00000000', ' Caption', ' lib', 'ETHERTYPE', 'velt', 'ESULT', 'oxic' |
| '..', ' ..', '..\', 'hers', ' DSL', 'GHz', ' VALUES', '.."', 'mic', ' Experiment' |

Table 1: When comparing the decoded top 5 right singular vectors in the memory head vs. the context head, we notice a clear trend in which the memory head especially encodes geography related information.

could be a result of the need for a more extended label field. The COUNTERFACT dataset includes broad label fields beyond geographical information such as names, dates and etc. The specific head we selected (15.7) is shown to encode memory in a specific field, therefore, this could lead to the poor performance in COUNTERFACT.

## 7 Discussion & Future Work

This paper investigates factors that influence a model's propensity to favor in-context vs. memorized factual associations, when the two compete with one another. Our results demonstrate that the frequency of information in the pretraining corpus can affect the model's tendency to use new, conflicting information provided in context. Building on this, we provide a proof of concept that this tendency can be controlled by a mechanism in the attention heads which allows us to manipulate LMs' tendency to prefer new in-context information without modifying any model parameters directly. By building off insights from mechanistic interpretability, we can localize single attention heads that contribute to this mechanism. This provides evidence that decomposing complex neural networks into understandable components is possible, even in models with billions of parameters. Still, we observe that the selected heads promote domain specific knowledge rather than a more abstract concept of truthfulness. This brittleness is characteristic of mechanistic analyses of larger models, and should be a priority for future work. Nonetheless, given the early stage of research on this level of analysis of large language models, findings of this type even in an isolated setting are exciting and can lay the groundwork for subsequently discovering more general mechanisms.

The exploratory methods described here suggest avenues via which future work might develop more sophisticated techniques for controlling and auditing deployed language models. Adapting LMs post-hoc for applications that require domain-specific information is a growing problem. For example, there are simultaneously reasons we might want to suppress the use of in-context information at run time (e.g., to combat prompt-injection attacks) as well as reasons we might want to encourage it (e.g., to enable users to provide new, personalized, or hypothetical information to the model). The intervention we describe in this work is intriguing in that it can be used without changing the model and can be turned on and off dynamically within the forward pass. It thus offers a promising direction for further work on model editing.

## 8 Conclusion

In the problem setting of predicting world capitals, our results show that the ability of language models (LMs) to overwrite information that it memorized in pretraining depends both on the frequency of the subject of the new fact (the country, e.g., Poland), as well as the frequency of the overwriting information (the counterfactual city, e.g., London). We can intervene on attention heads that we find tend to push the prediction one way or another. By simply rescaling the value vectors of important heads, we can control which city the model predicts without updating any model parameters. We hope these results encourage future work in understanding the internal structure of neural networks in general.

## Limitations

Our work aims to show that individual components in LMs can play predictable roles in certain model behaviors; in this case, whether or not to overwrite memorized information about world capitals. Further work is required to understand how to control the use of context or memorized information in generated text for this to be successfully applied in the most general cases. The dataset we use is templated and applied to the limited domain of country-capital relationships, meaning that we can not make general statements about the role of individual attention heads in arbitrary context. It is likely, given the flexibility of LMs, that many different components can play this role depending on the nature of the task. This work contributes to the growing body of evidence that individual components (e.g., attention heads) *can* specialize for certain roles across contexts. We can not yet show how to control this behavior in arbitrary settings, but we provide a promising avenue for how this might be done in the future.

## 9   Acknowledgments

We thank Catherine Chen, William Rudman, Charlie Lovering, Apoorv Khandelwal, Michael Lepori, Louis Castricato, Samuel Musker, Aaron Traylor, Tian Yun for discussion and comments on this work. We thank Daniel Wong for providing hardware support during writing.

## Ethics Statement

Our work provides early indicators of future tools that could aid in making models safer and more trustworthy. Insights like those described could potentially lead to methods for better predicting how language models might behave in certain settings, preventing models from generating personal information learned in pretraining (preventing access to some memorized information), or the opposite, preventing prompt injections from affecting model behavior (preventing access to certain context information). Although we do not observe the quality of generated text changing substantially in our limited setting, future work is needed to better understand how manipulating the 'intensity' of model components, especially those which affect the recall of pretraining information, can alter model behavior or make it easier to extract memorized text containing personal information.

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

## A  Effect of Term Frequency on Additional Models

In the main paper, we mainly focus on the term frequency effect on the in-context and memorized predictions on Pythia-1.4b. We applied the same analysis on the the rest of the Pythia models and series of GPT2 models: Pythia-70m, Pythia-160m, Pythia-410m, Pythia-1b, Pythia-2.8b, GPT2, GPT2-medium, GPT2-large, GPT2-xl. We observe the similar conclusions as Section 5. Across all the models, there is a cohesive trend of blue line going down as the frequency increases and pink link going up. The same conclusion align: as the frequency goes up, models are more likely to predict the memorized answers. As the frequency decrease, models prefer the in-context answers. See figures 8, 9, 11, 10, 12,13, 14, 15, 16

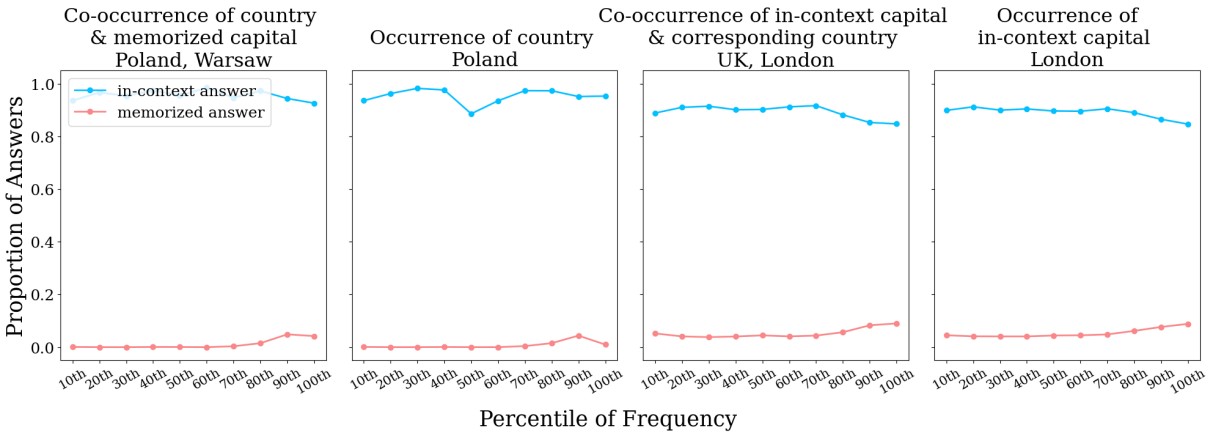

Figure 8: Frequency Effect on GPT2

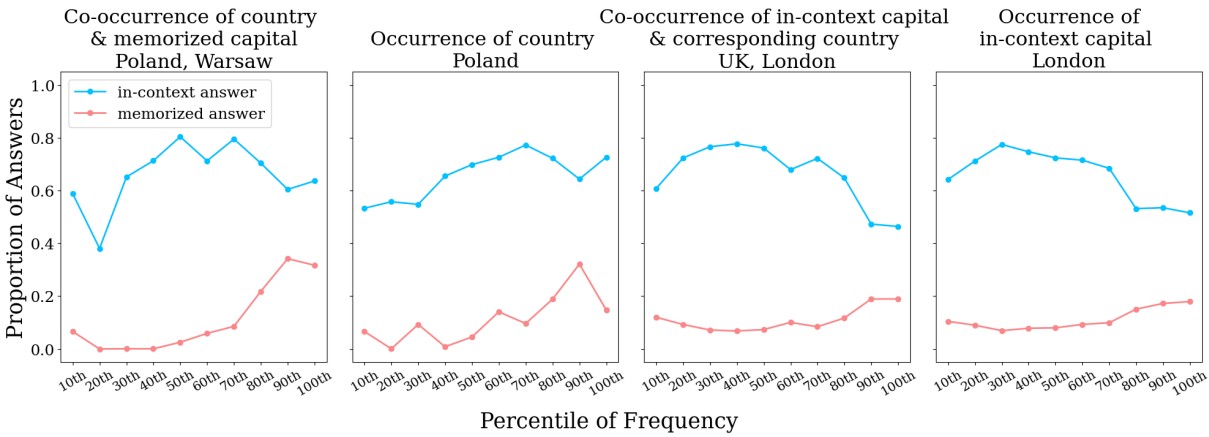

Figure 9: Frequency Effect on GPT2-medium

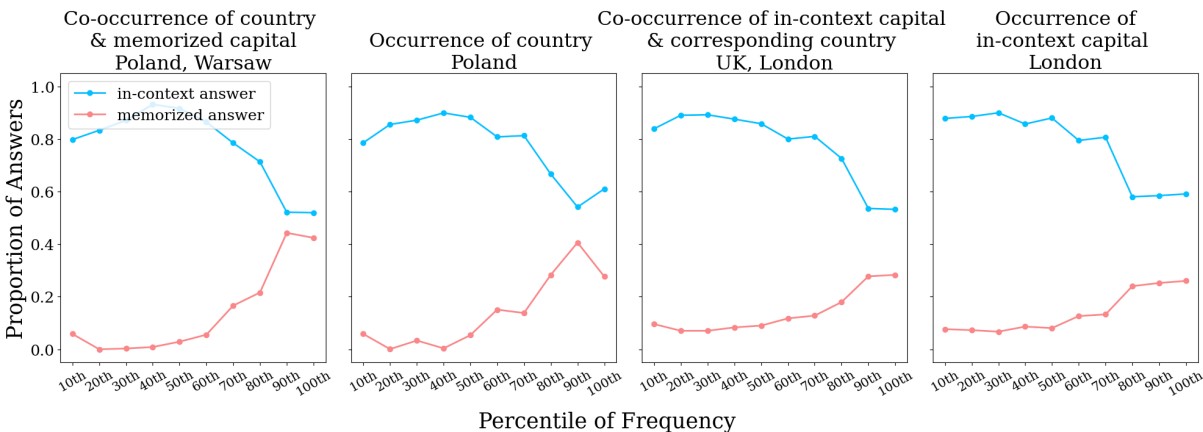

Figure 10: Frequency Effect on GPT2-large

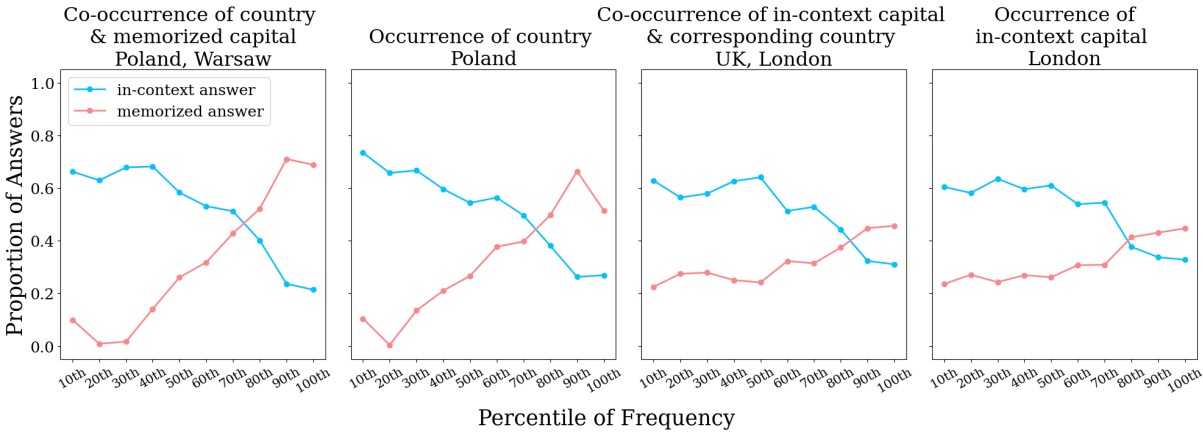

Figure 11: Frequency Effect on GPT2-xl

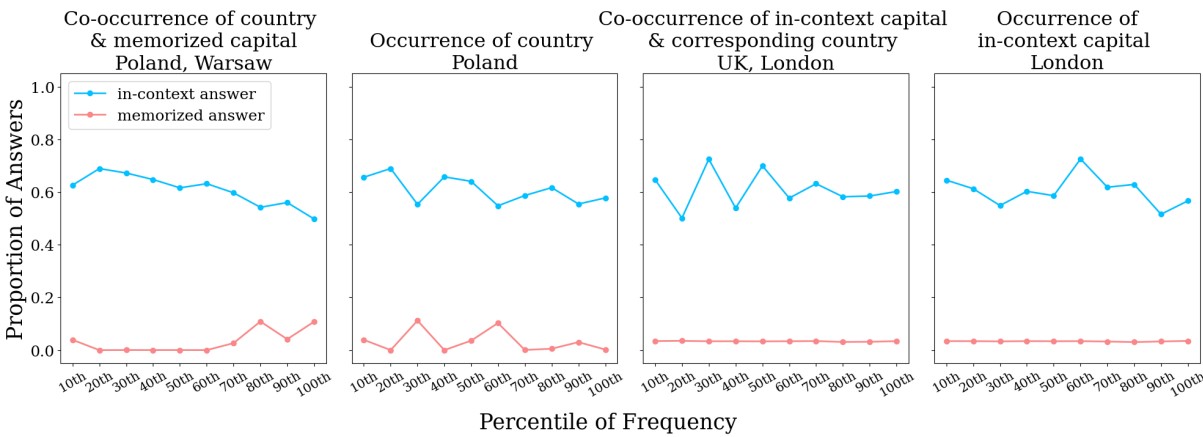

Figure 12: Frequency Effect on Pythia-70m

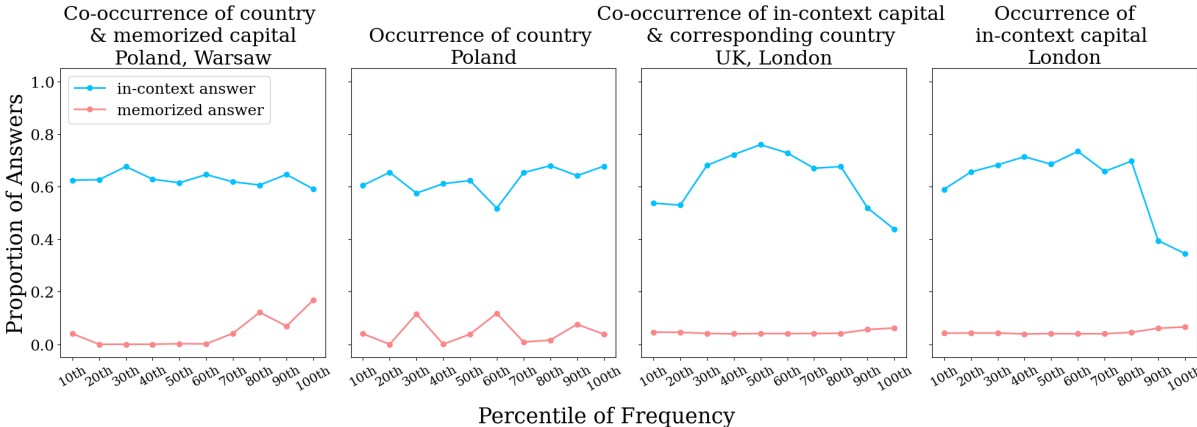

Figure 13: Frequency Effect on Pythia-160m

Pythia-410m Frequency Effect on In-Context and Memorized Answers Predictions

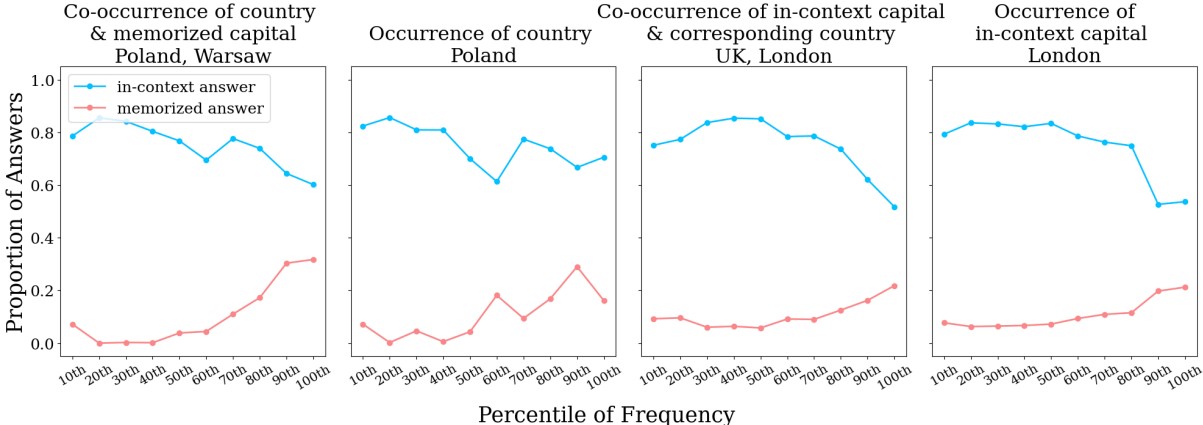

Figure 14: Frequency Effect on Pythia-410m

Pythia-1b Frequency Effect on In-Context and Memorized Answers Predictions

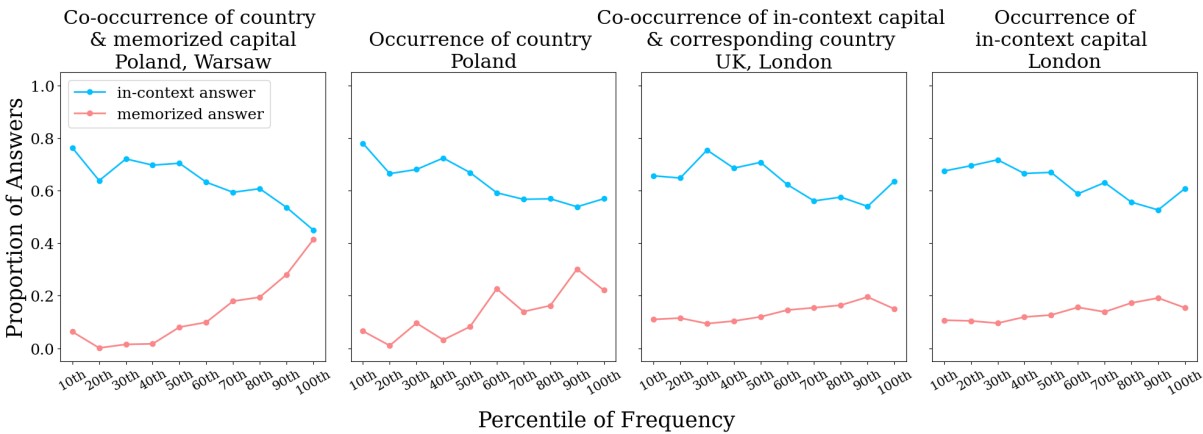

Figure 15: Frequency Effect on Pythia-1bm

Pythia-2.8b Frequency Effect on In-Context and Memorized Answers Predictions

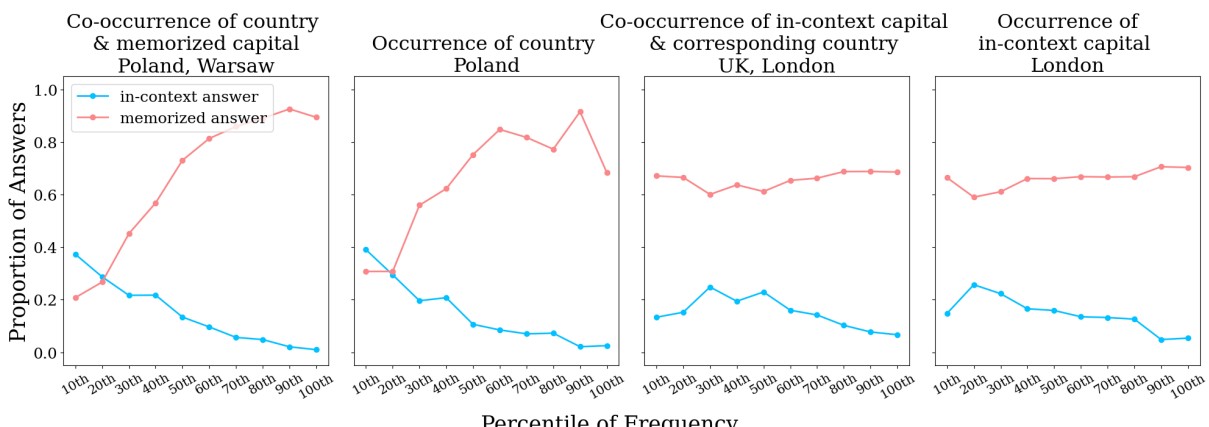

Figure 16: Frequency Effect on Pythia-2.8b

## B  Effect of Model Size

Here, we show the effect of model size in relation to all of the measures of term frequency including those which we did not include in the main paper. This includes frequency with respect to the memorized country, in-context capital, co-occurrences of the memorized country and capital, and the co-occurrences of the in-context capital and its corresponding country.

### B.1  Pythia Series

See figures 17, 18, 19, 20

### B.2  GPT2 Series

See figures 21, 22, 23, 24.

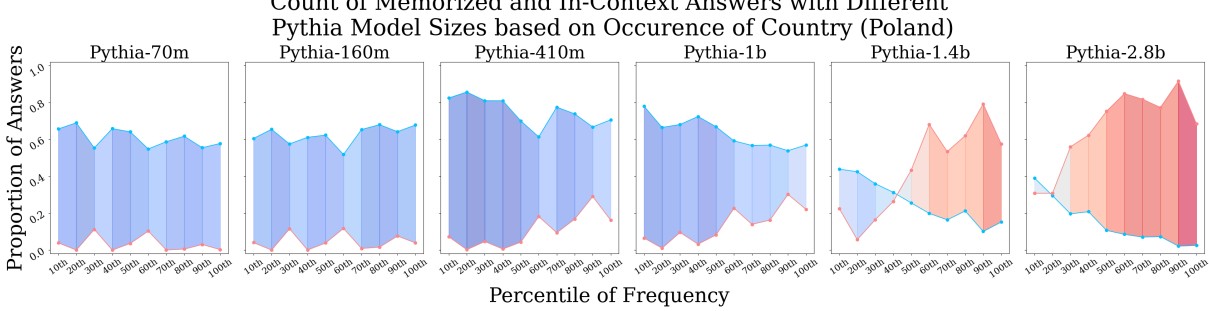

Figure 17: Pythia-Effect of model sizes on predictions based on the frequency of *country*

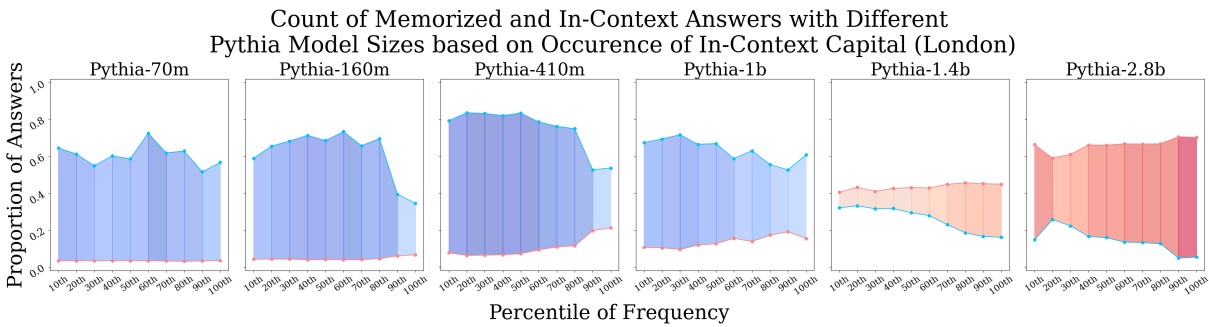

Figure 18: Pythia-Effect of model sizes on predictions based on the frequency of in-context capital

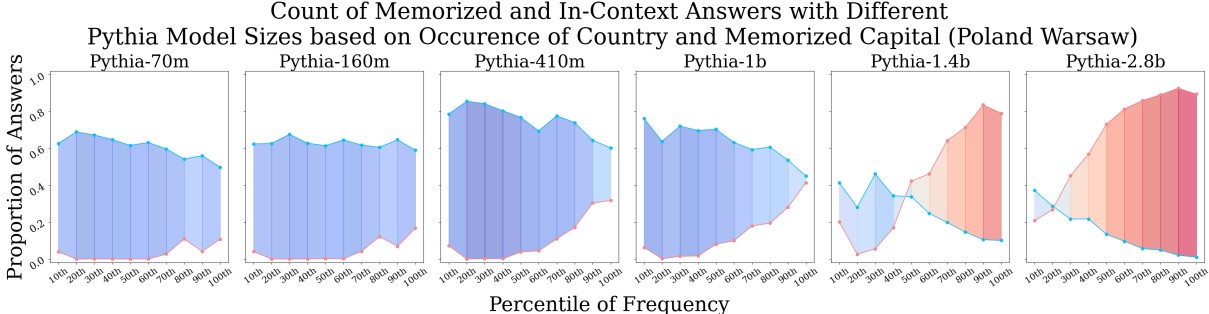

Figure 19: Pythia-Effect of model sizes on predictions based on the frequency of *country* and memorized capital

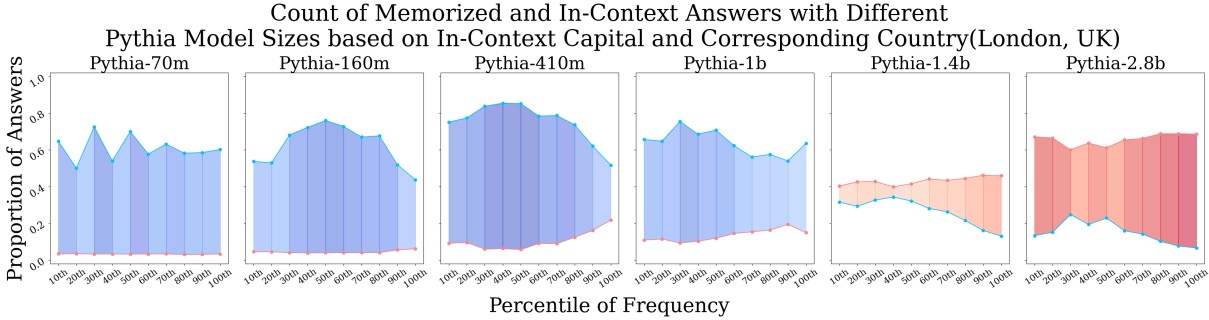

Figure 20: Pythia-Effect of model sizes on predictions based on the frequency of in-context capital and corresponding country

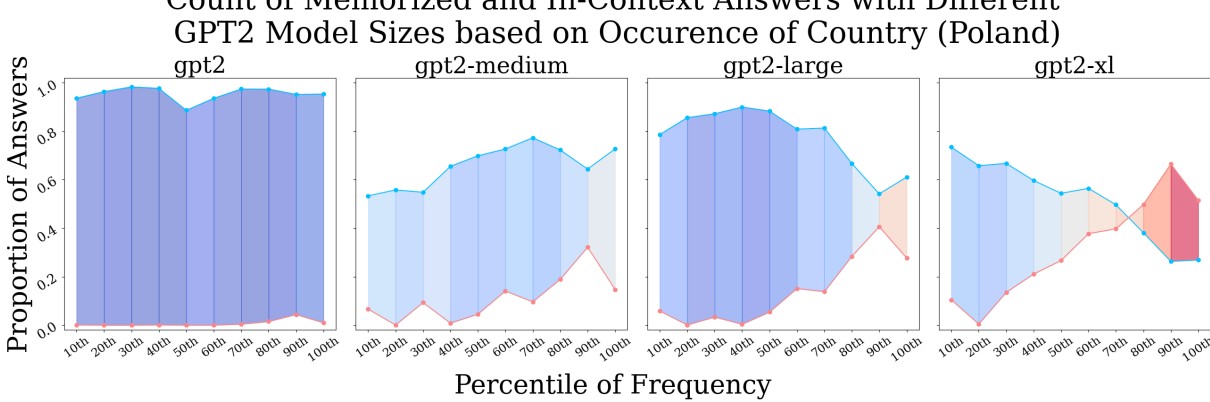

Figure 21: GPT-2 Effect of model sizes on predictions based on the frequency of *country*

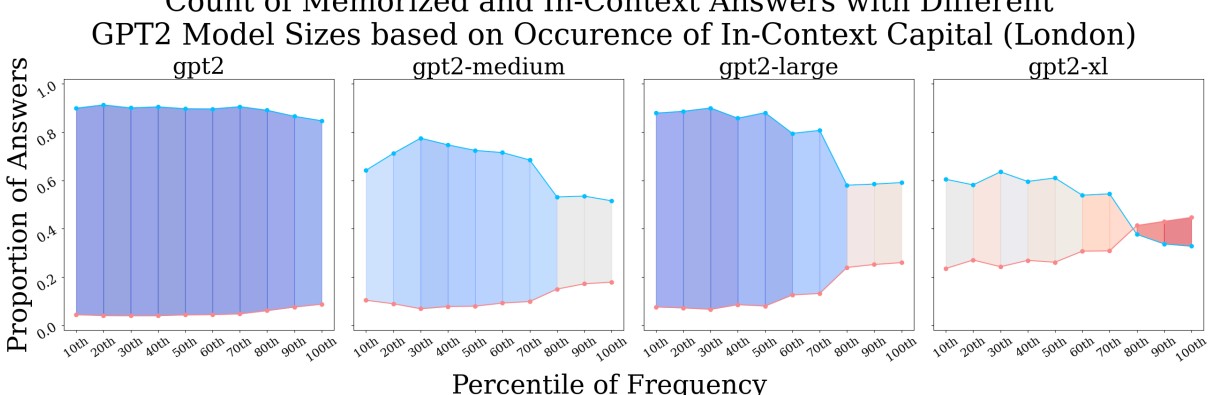

Figure 22: GPT-2 Effect of model sizes on predictions based on the frequency of in-context capital

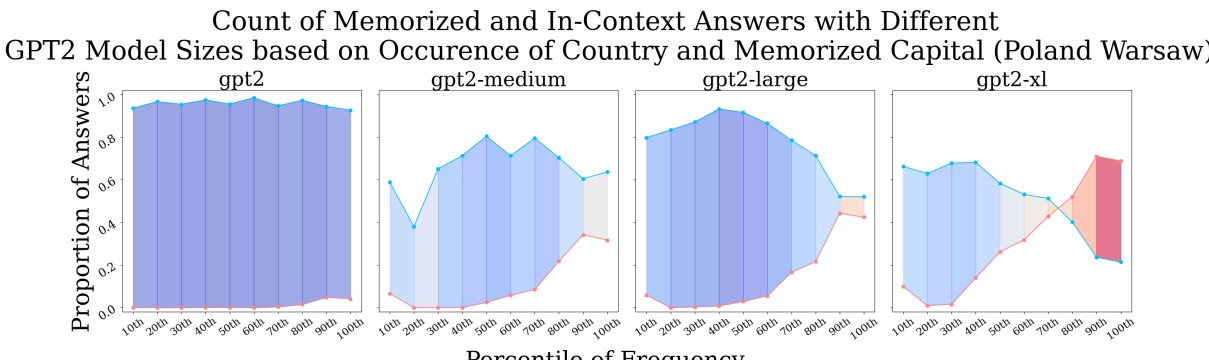

Figure 23: GPT-2 Effect of model sizes on predictions based on the frequency of *country* and memorized capital

Count of Memorized and In-Context Answers with Different
GPT2 Model Sizes based on In-Context Capital and Corresponding Country (London, UK)

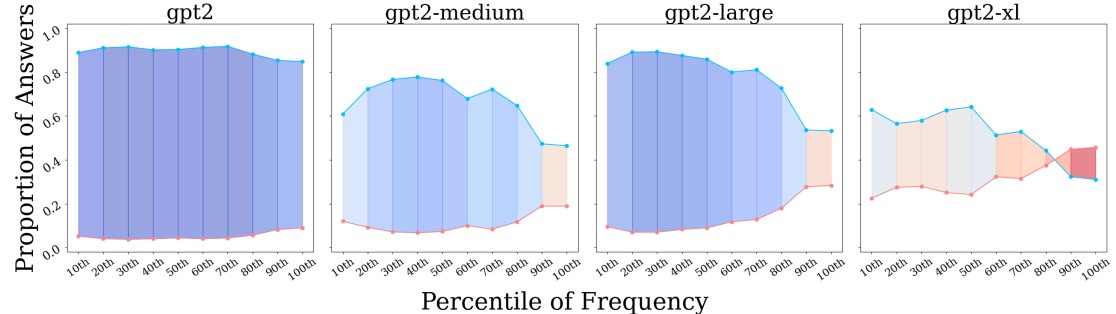

Figure 24: GPT-2 Effect of model sizes on predictions based on the frequency of in-context capital and corresponding country

| Model | Memory Head | In-Context Head |
|---|---|---|
| *Pythia-1.4b* | 15.7 | 19.12 |
| *Pythia-2.8b* | 17.17 | 17.31 |
| *gpt2-xl* | 35.19 | 25.20 |

Table 2: Specific memory head and in-context head

## C   Head Attribution

Attention heads are vectors of size $d_{head} = d_{model}/n_h$ where $n_h$ is the number of heads in the model. The standard way to compute the output of attention layers is by concatenating all of the heads to form one $d_{model}$ sized vector, and passing it through an output weight matrix $W_O^H$ of size $(d_{model}, d_{model})$. This would initially seem to prevent us from directly projecting an individual head from $d_{model}$ space into the vocab space using the unembedding matrix, since to get to $d_{model}$ space, each head has to interact in the $W_O^H$ multiplication. However, as observed by (Elhage et al., 2021), this operation is equivalent to splitting the $W_O^H$ matrix into $n_h$ $(d_{head}, d_{model})$ sized chunks, projecting individual heads and adding the projections up. If we consider the output of an attention layer by stacking the attention result vector $r^{h_1}, r^{h_2}, ...$ and multiply by an output matrix $W_O^H$, we can split $W_O^H$ into each size blocks for each heads $[W_O^{h_1}, W_O^{h_2}...]$. Therefore, we can get the contribution to the attention output for every head via $W_O^{h_1} \dot{r}^{h_1}$.

### C.1   Memory Head & In-Context Head

### C.2   Individual Heatmap

See Figure 25

### C.3   Tuning Scale

In the main text, we show the tuning scale for Pythia 1.4B. Here we present tuning curves for other models on their corresponding memory and in-context heads. See figures 26,27

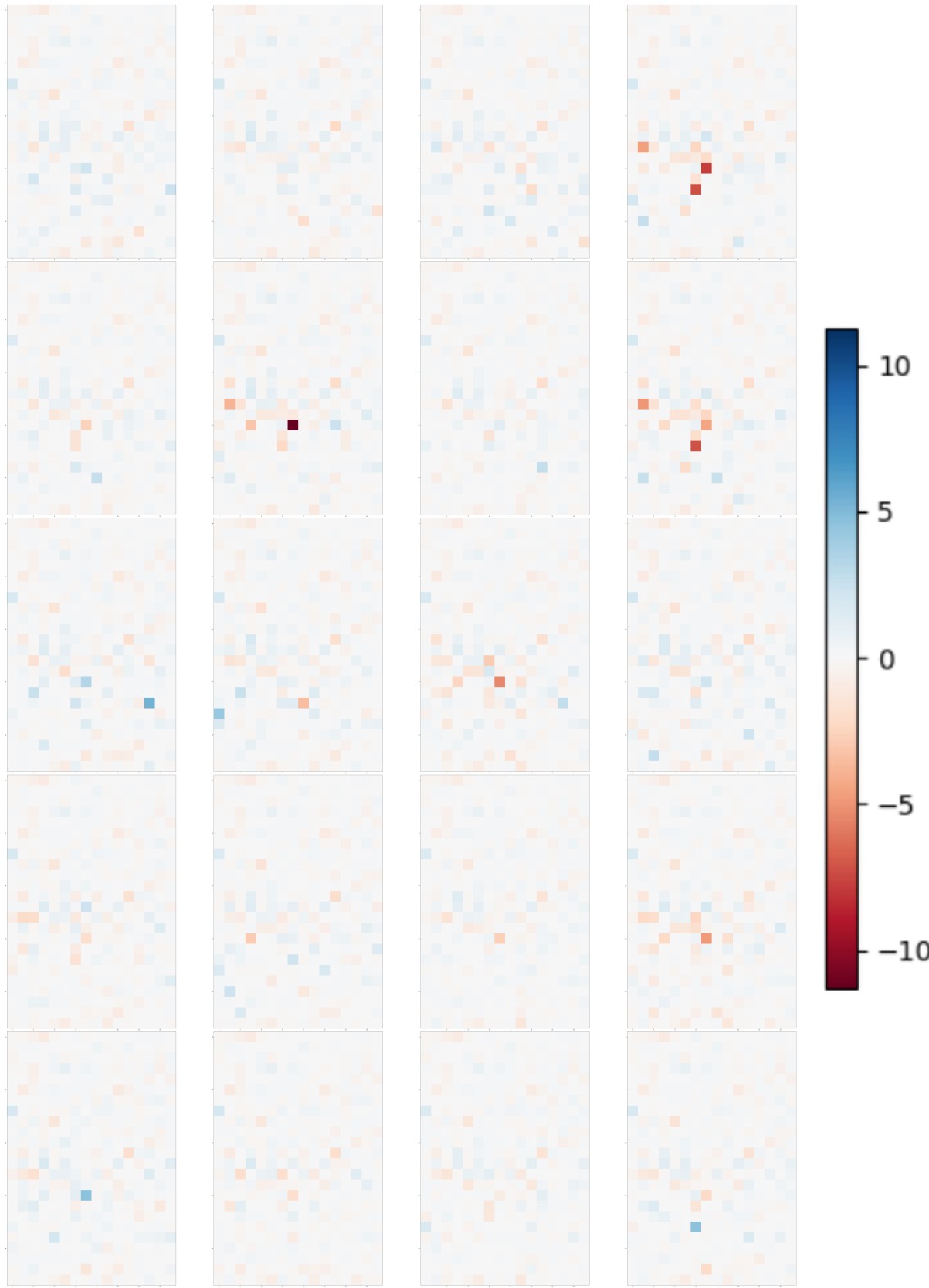

Figure 25: We extract individual heatmaps of head attribution across 20 randomly selected examples. We locate the maximum positive values (4) and the the minimum negative value (-10). The scale is set from -10 to 10 to ensure the same domain of all the maps. We can observe that there are more prominent memory head (dark red) compared to in-context head (dark blue). We can also see that the distribution of the memory heads and the in-context heads vary across different examples.

Pythia-2.8b Tuning Scale

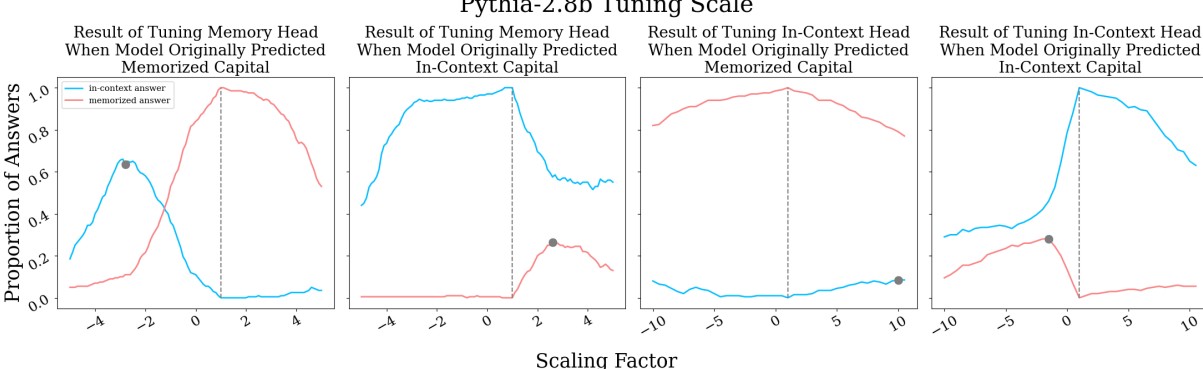

Figure 26: Pythia-2.8b Tuning Scale

GPT2-xl Tuning Scale

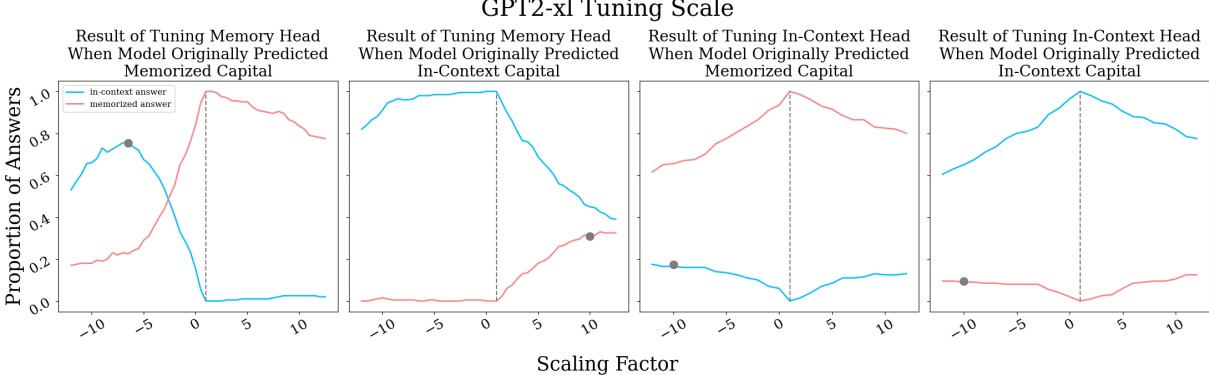

Figure 27: GPT2-xl Tuning Scale

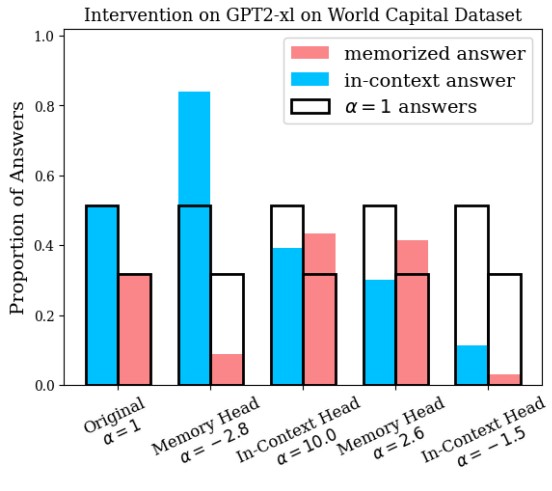

Figure 28

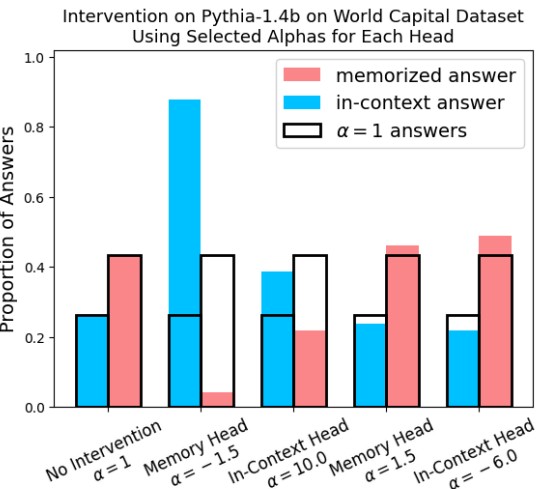

Figure 29

## D  Interventions

Here, we provide the intervention experiments that we could not include in the main text.

### D.1  World Capital Dataset

See figures 28, 29, 30

### D.2  COUNTERFACT Dataset

See figures 31, 32, 33

### D.3  Frequency Effect from Intervention

See figures 34, 35, 36, 37, 38, 39, 40, 41,42, 43, 44, 45

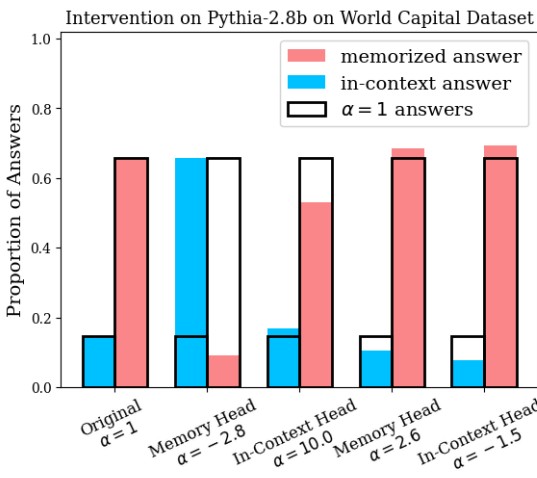

Figure 30

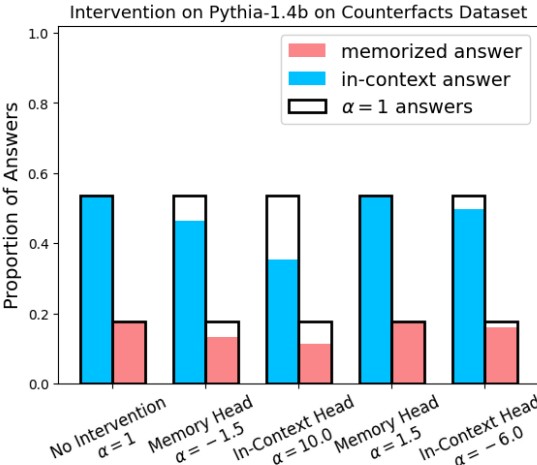

Figure 31

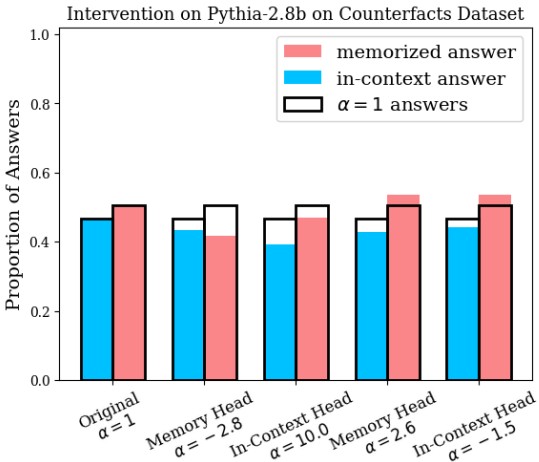

Figure 32

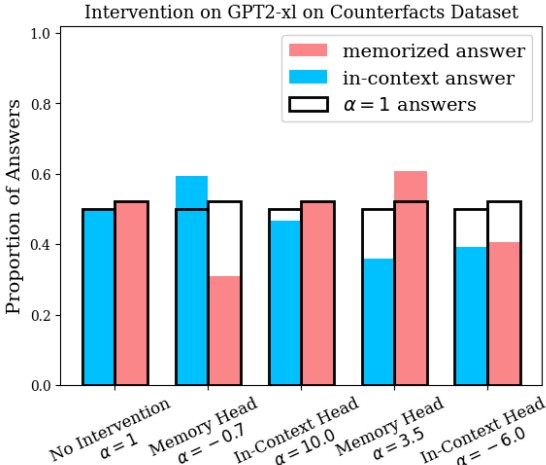

Figure 33

## Pythia-1.4b Intervention on Memory Head 15.7 with -1.5 Scale

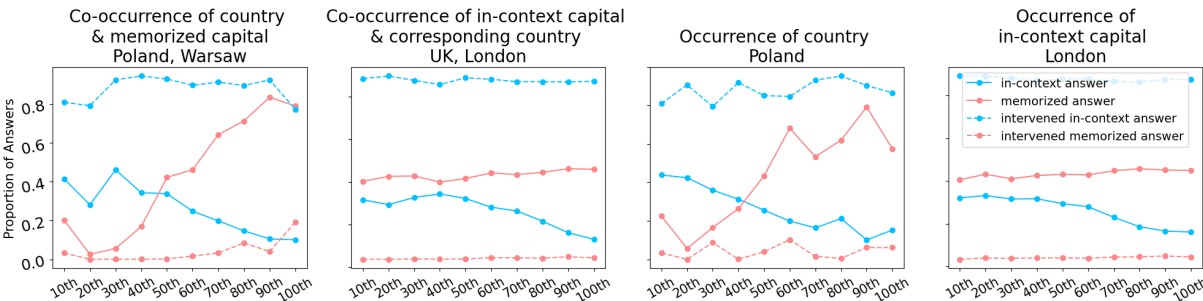

Figure 34

## Pythia-1.4b Intervention on Memory Head 15.7 with 1.5 Scale

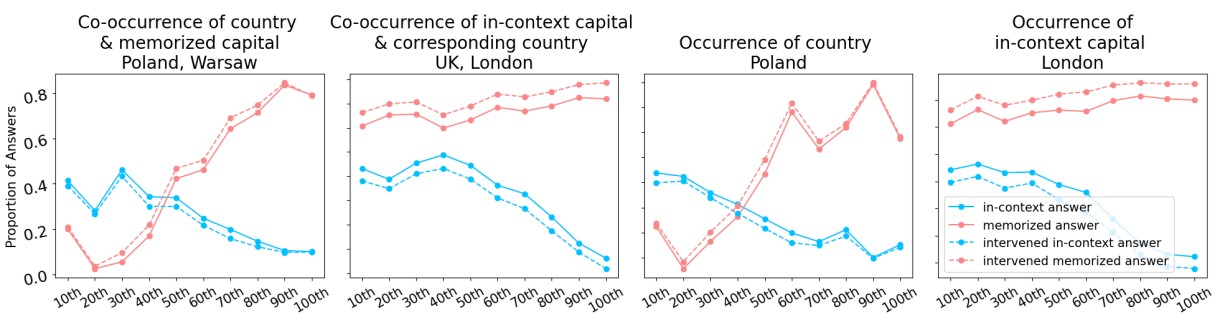

Figure 35

## Pythia-1.4b Intervention on In-Context Head 19.14 with -6 Scale

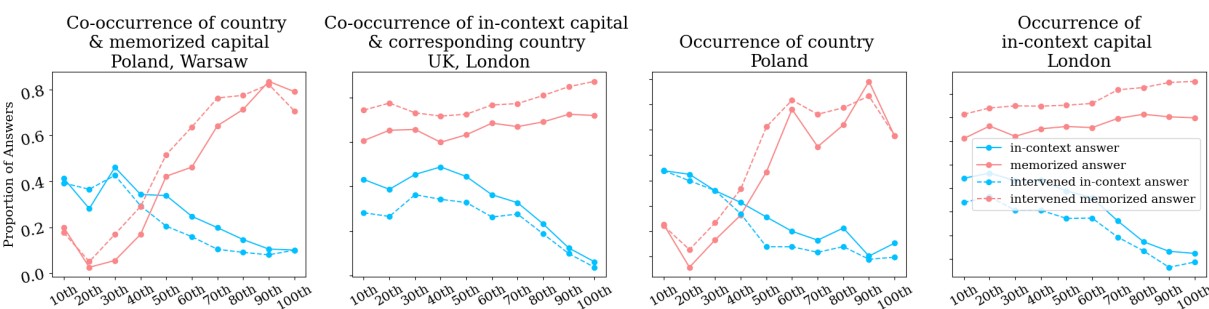

Figure 36

## Pythia-1.4b Intervention on In-Context Head 19.14 with 10.0 Scale

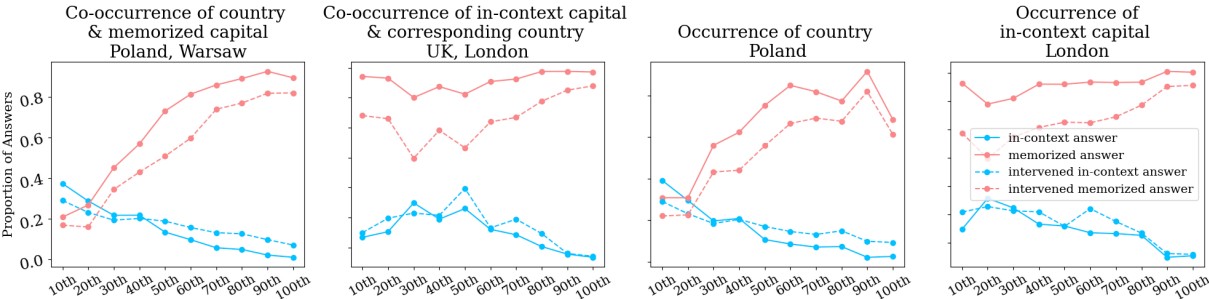

Figure 37

## Pythia-2.8b Intervention on Memory Head 17, 17 with -2.8 Scale

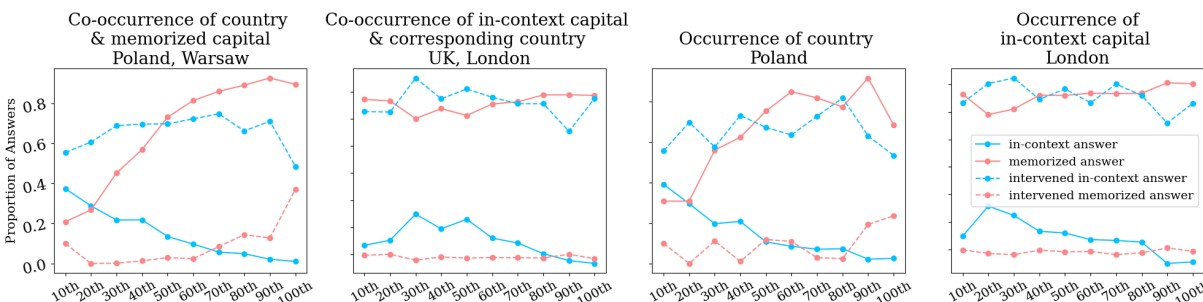

Figure 38

## Pythia-2.8b Intervention on Memory Head 17, 17 with 2.6 Scale

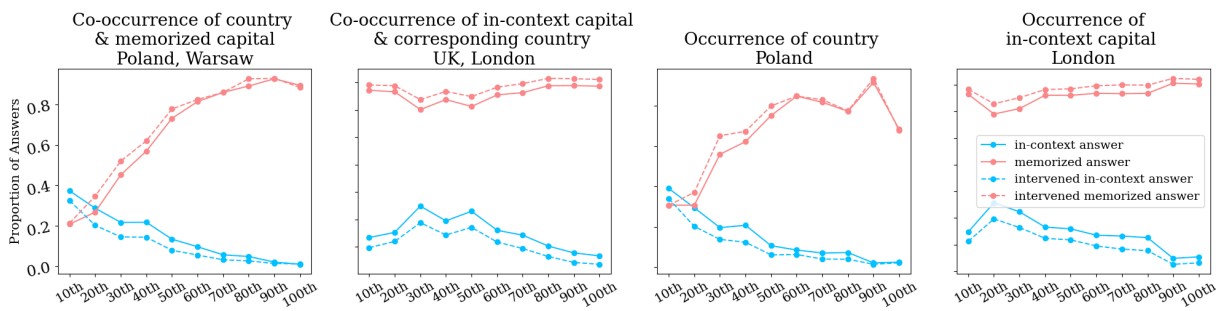

Figure 39

## Pythia-2.8b Intervention on In-Context Head with -1.5 Scale

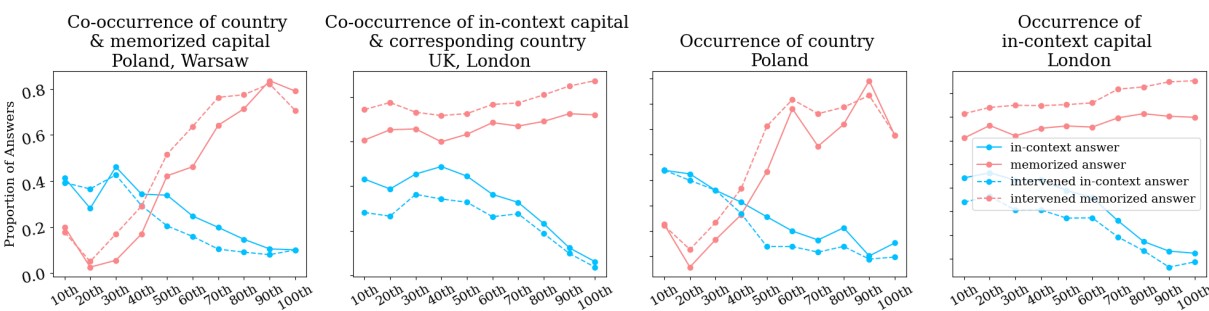

Figure 40

## Pythia-2.8b Intervention on In-Context Head with 10 Scale

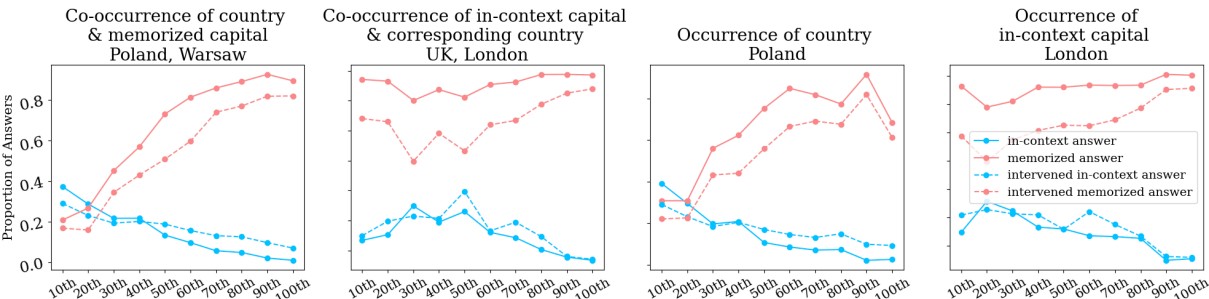

Figure 41

## GPT2-xl Intervention on In-Context Head 29, 20 with -10 Scale

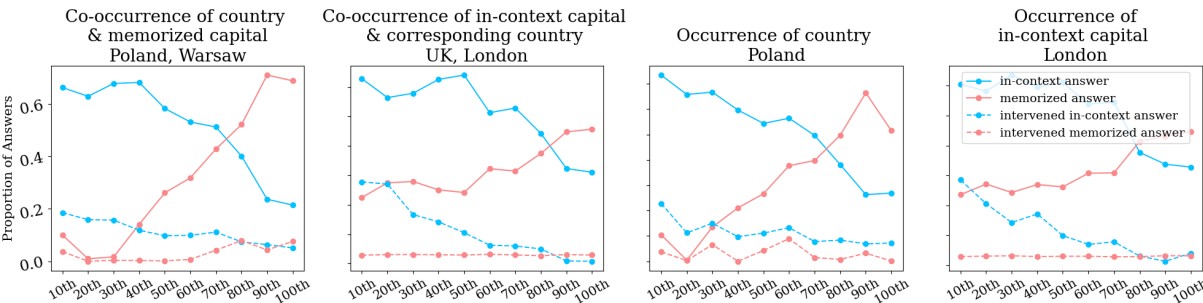

Figure 42

## GPT2-xl Intervention on Memory Head 35, 19 with 10 Scale

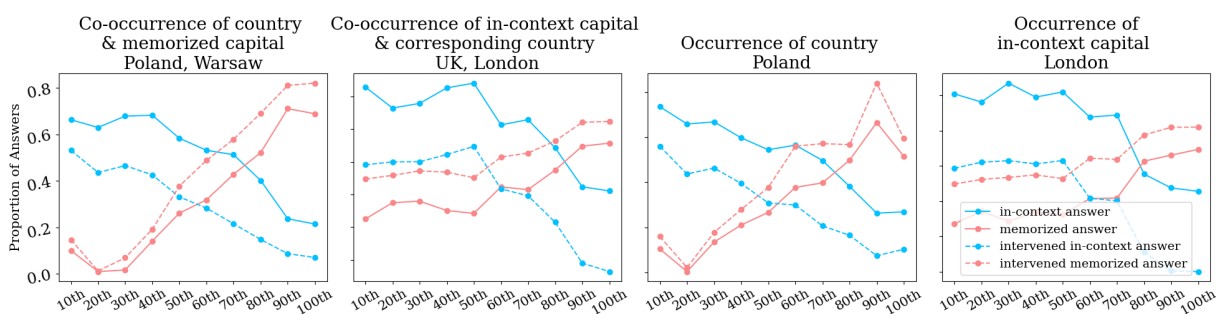

Figure 43

## GPT2-xl Intervention on Memory Head 35, 19 with -6.8 Scale

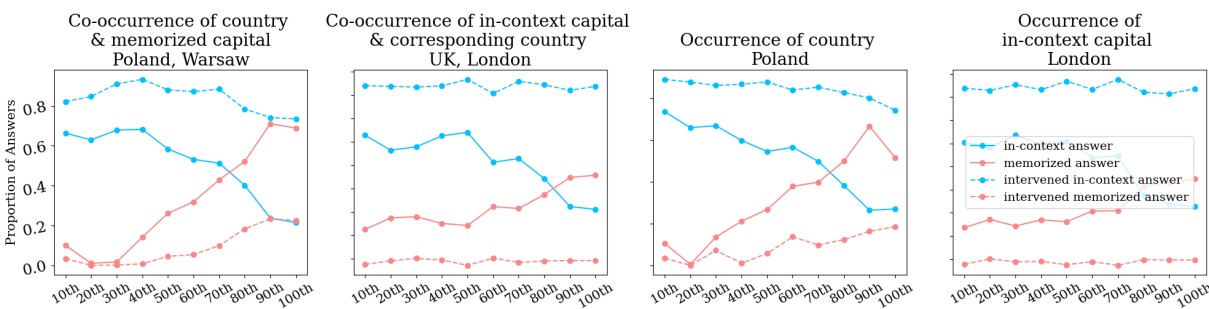

Figure 44

## GPT2-xl Intervention on In-Context Head 29, 20 with 10 Scale

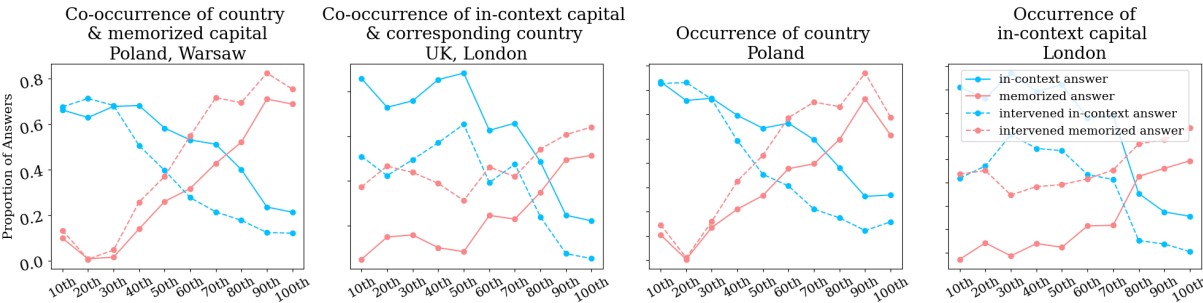

Figure 45

## E  Decoded Result Analysis

Here we look into specific decoded result before and after analysis to see how does the intervention change the decoding. We focus on Pythia-1.4b for this analysis. In the following example,

```
The capital of Antigua and Barbuda is
Tirana.
Q: What is the capital of Antigua and
Barbuda?
A:
```

The original decoded text before intervention is:

```
The capital of Antigua and Barbuda is
Tirana
Q: What is the capital of Antigua and
Barbuda?
A: Antigua and Barbuda is the capital
of Antigua and Barbuda
```

After the intervention by negatively tuning the memory head,

```
The capital of Antigua and Barbuda is
Tirana
Q: What is the capital of Antigua and
Barbuda?
A: The capital of Antigua and Barbuda
is Tirana.
```

We observe that after the negative tuning the memory head, one of the biggest change in the decoded text is that increase of decode text begin with `The capital of <country> is`. In Pythia-1.4b, before the negatively tuning the memory head, only 7961 decoded test begins with `The capital of <country> is`. After the intervention, this number raise to 57440. More than 90% of the predicted answers begins with `The capital of <country> is`. This change significantly prompt the increase the in-context. 92% of the text beginning with `The capital of <country> is` predicts in-context answer. By copying the injected prompt, the model are able to predict the in-context answer. We hypothesize that tuning down the memory head will help the model to pay more attention to the given prompts and implement the copying task. More experiments are required to test this hypothesis.

Moreover, we also observe that a common reason for model to predict neither the in-context answer nor the memorized answer that the model will just repeat itself by outputting the given country name. In the above example, the model output *A: Antigua and Barbuda is the capital of Antigua and Barbuda* before intervention. 77% of the decoded answer that gets neither the in-context answer nor the memorized answer simply just repeat the given country name in context. 83% of these answer are changed to predict the memorized answers after negatively tuning the memory head. Negatively tuning the memory head can be responsible for shifting the copying mechanism from the country name ahead to the in-context capital.

## F    Head 11.11 Analysis

In another set of sample, we find head 11.11 that have the similar effect as the chosen head 15.7. However, we didn't include this head in the main paper since this head offers less interpretable explanation. See figures 46, 47, 48, 49, 50

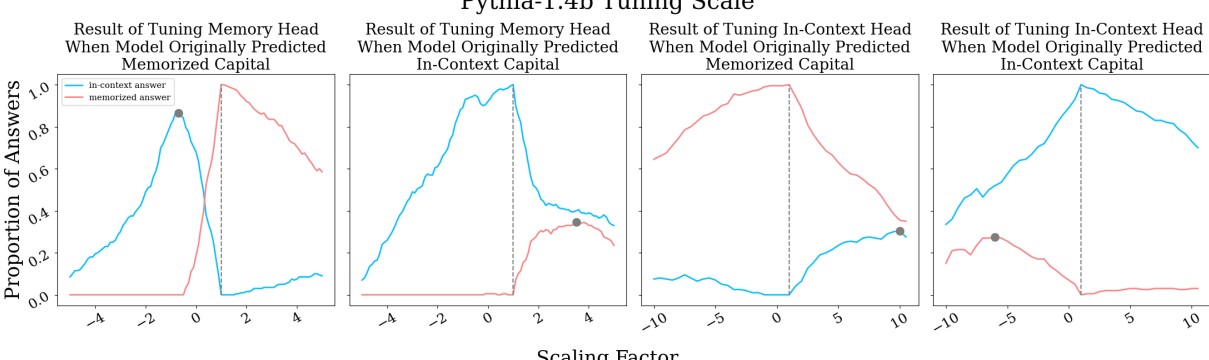

Figure 46: Scale graph on intervention for head 11.11

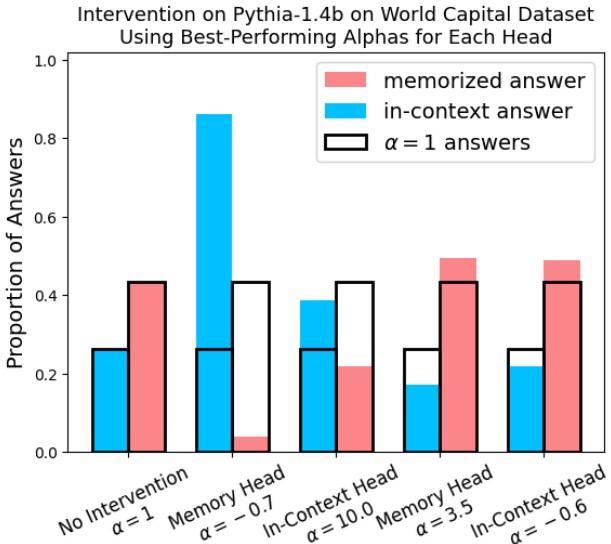

Figure 47: Overwrite result with respective scale on Head 11.11 on the world-capital dataset

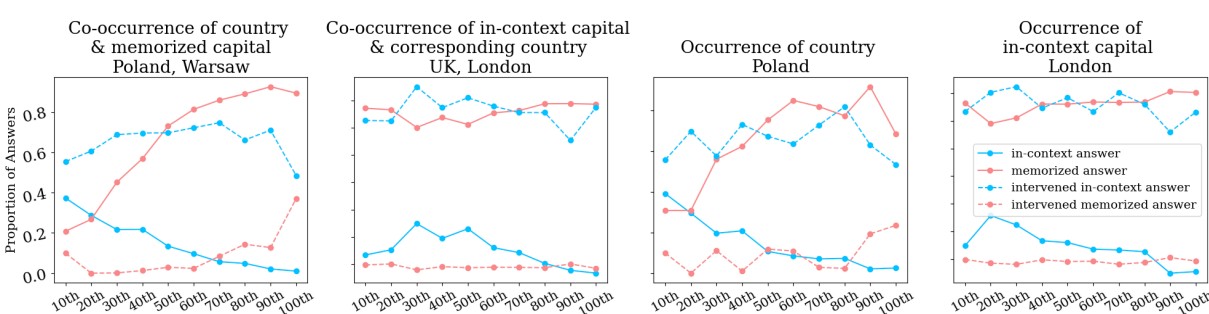

Figure 48: frequency effect on negatively tuning head 11.11

## Pythia-1.4b Intervention on Memory Head 11.11 with 3.5 Scale

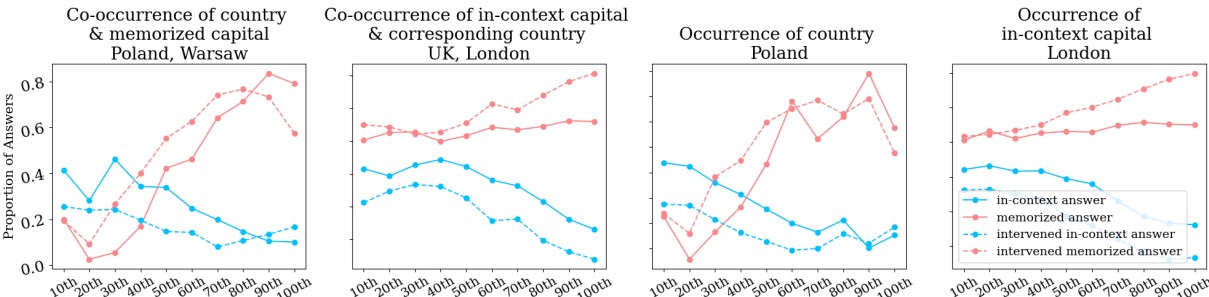

Figure 49: frequency effect on positively tuning head 11.1

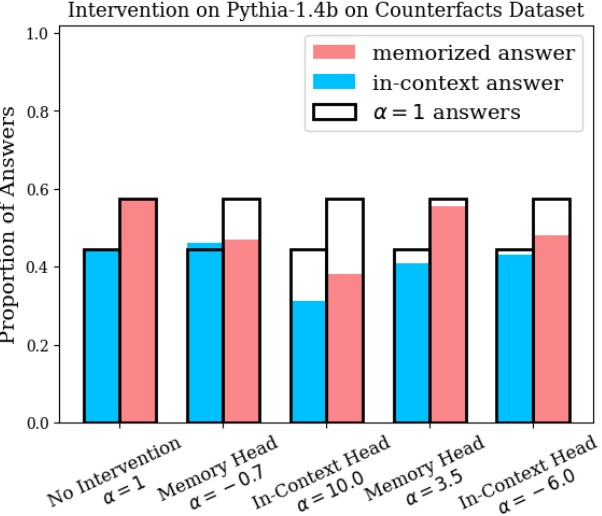

Figure 50: Overwrite result with respective scale on Head 11.11 on the COUNTERFACT dataset

# G  Singular Value Decomposition

We included the top 10 decode for all 64 right singular vectors in head 15.7 and head 19.4 in the main paper to show that the memory head weights specifically encourage geography related terms in the next token prediction. 15.7 shows interpretable decode clusters around the location information. However, head 19.4 didn't show any meaningful clusters. We include the decodings for each of the singular vectors in each head in this section, see Tables 3 4.

| | | | | | | | | | |
|---|---|---|---|---|---|---|---|---|---|
| ' LW', ' Wade', ' WI', 'liche', 'ienne', ' ell', 'owe', 'iale', 'uelle', '\u0435 \u0442 \u0435' |
| ' Italian', 'Italian', ' Italy', ' Ital', ' Io', ' Giovanni', ' pasta', 'Io', ' Giul', ' Naples' |
| 'WA', 'WS', ' WA', 'owa', 'ws', 'wa', 'Ws', ' Wa', 'pora', ' WI' |
| ' WM', 'WM', 'wm', 'mw', 'w', 'nw', ' w', ' MW', ' Minnesota', 'WN' |
| ' Guatemala', ' Guatem', 'usta', 'osta', ' Tampa', ' Brazil', 'ativa', ' Bah', ' Tamil', 'Brazil' |
| 'Greek', ' Greek', ' Greece', ' Knox', ' Greeks', 'kappa', ' Athens', 'greek', ' Kim', ' Athen' |
| ' Ji', ' MN', ' NJ', ' Swiss', ' Jiang', ' Peng', 'Italian', ' Italian', 'azz', 'elli' |
| ' Kansas', ' NH', 'NH', ' Nebraska', ' Omaha', ' NZ', 'maha', 'nek', ' Nepal', ' Het' |
| ' Brazilian', ' Og', 'Brazil', ' Brazil', ' Nigerian', ' Ethiop', ' Brasil', ' Nigeria', ' Danish', ' Ethiopia' |
| 'KM', ' KL', 'mM', 'Kam', 'KD', ' HK', 'mw', 'MW', ' LM', 'KH' |
| 'BN', ' BN', ' HK', ' ku', ' Uk', 'Uk', ' RN', ' Yuk', ' Nak', 'HK' |
| ' Kaz', ' mos', ' Raz', ' KL', ' Jed', ' Malaysian', ' Malay', ' Sultan', ' Kom', ' KK' |
| 'MK', ' Panthers', ' Argentine', ' Spart', ' Carolina', 'AZ', ' MK', 'South', 'zc', ' Chile' |
| ' Moz', ' OM', ' Miz', ' Sach', ' MG', 'XM', ' O', 'OWS', 'OMN', ' Judaism' |
| ' NK', ' Kore', ' Milwaukee', ' ND', ' NL', 'KN', ' Korea', ' Norway', 'NL', ' Koh' |
| 'gh', ' Henderson', 'Gh', 'nh', 'vh', ' Dh', 'GH', 'dh', ' GH', ' Gh' |
| ' Rio', ' Trin', ' Mississippi', 'AO', ' Munich', 'Mb', 'vor', 'BV', '\u00F2', 'Brazil' |
| ' Filip', ' Mexican', ' Philippines', 'lv', ' Manila', ' VA', 'NV', ' NV', ' Philippine', ' ANC' |
| 'KP', ' KDE', ' Hung', 'DK', 'Hung', 'KR', 'Viet', ' Kai', ' KD', 'asian' |
| 'HT', ' Hos', 'EH', ' HT', ' Texans', ' Haw', 'HA', ' Texas', 'UH', ' HI' |
| 'w', 'wi', ' Wu', ' w', ' Wa', ' Kai', 'wu', 'WA', 'nw', ' WA' |
| ' Malta', ' Miami', 'Dutch', ' Midd', ' Dutch', ' Amsterdam', ' Caribbean', ' Netherlands', 'mf', ' Jamaica' |
| ' li', ' l', ' LI', ' Li', ' lis', ' LA', 'LL', 'lb', ' LN', 'LB' |
| ' Kumar', ' Indianapolis', ' Krish', 'ji', ' Mari', ' Birmingham', ' Kansas', ' MG', 'IU', ' Indy' |
| ' Abdullah', ' Alger', 'FN', ' FN', 'bn', ' Saudi', ' Ahmed', ' BN', ' Niger', ' Belgium' |
| ' Ohio', 'VT', ' OH', 'Ohio', ' Wisconsin', ' Volkswagen', ' VT', '\u0442 \u0435', 'vt', ' Hamilton' |
| ' Indones', ' Indonesian', ' Indonesia', ' Ramos', ' Lopez', ' Flores', 'Dutch', ' Java', ' Luxem', ' Jorge' |
| 'wc', 'RPC', ' Va', ' Richmond', 'WS', ' Dou', ' CR', ' Fran\u00E7ois', 'qs', ' WS' |
| ' Trent', ' TT', ' Gomez', ' Scotia', 'TT', ' Dalton', ' Thom', 'Tg', 'ammar', ' Iceland' |
| ' Montana', ' Alaska', 'sdk', ' Plains', ' Norway', ' Sau', ' FX', ' Kashmir', 'Pak', ' Mong' |
| ' Sanchez', 'Mi', ' Fernando', ' Miami', 'SI', 'Indian', ' Si', ' Sar', ' S\u00E3o', ' IB' |
| ' Japan', ' Japanese', ' Singapore', 'Japanese', 'Japan', ' Taiwan', 'jp', 'jn', ' Tokyo', ' Lee' |
| ' Kurd', ' Iraqi', ' EF', ' Ion', 'acci', ' Hond', ' Tex', ' Ecuador', ' Texas', ' Iraq' |
| ' AM', ' Abe', 'AMD', ' Am', 'MV', ' AMD', 'AM', 'amber', 'VA', ' AMP' |
| ' BV', 'BV', ' Eb', ' Kab', 'kB', 'BH', 'bv', ' vess', 'bbe', 'VB' |
| 'oux', ' Iowa', ' Sz', ' Shelby', ' Memphis', 'ocy', ' Saskatchewan', ' Ottawa', ' Sask', 'Sz' |
| ' Lah', 'LF', ' Nass', 'fel', 'lf', ' Levy', ' Nottingham', ' LF', ' FN', ' TN' |
| ' Hamburg', ' Texans', ' Houston', ' Tex', 'Houston', 'WL', ' TEXAS', 'gia', 'GAL', 'Gi' |
| 'HF', ' Holl', ' Hockey', 'hf', ' HF', ' Argent', 'FH', ' Argentina', ' HG', ' HM' |
| ' ETH', ' Seth', ' SG', ' iodine', ' Eph', ' Belfast', 'ETH', 'GS', ' Ish', 'IE' |
| 'French', ' Bav', ' French', ' Gust', ' w', ' Bou', ' franc', 'EG', 'Ot', ' TG' |
| 'OE', ' PF', ' PE', ' fp', 'opf', ' PO', 'EP', ' PDE', ' EP', ' Porter' |
| ' Rag', ' Maur', ' Dh', ' RCC', ' Karn', 'Mah', 'uid', ' Kal', ' Rodrig', ' Mah' |
| 'Lau', ' l', 'EC', ' Laurent', ' fro', 'Au', ' Tigers', 'Indian', 'chen', ' Lac' |
| 'Hay', 'hay', ' Chang', ' Hay', ' Bulls', 'hl', 'Rh', ' Kosovo', ' epiderm', ' SCC' |
| ' UC', 'UC', ' U', 'uca', ' ud', ' u', ' Sacramento', ' UP', ' uc', ' UDP' |
| 'QS', 'HS', ' Ecuador', 'qs', 'Sb', ' HS', 'WS', ' SES', 'oS', 'ns' |

| | | | | | | | | |
|---|---|---|---|---|---|---|---|---|---|
| ' Ig', | 'PQ', | ' Slav', | ' Iz', | 'CPP', | ' Pav', | ' Shapiro', | 'hens', | 'IQ', | 'RCC' |
| 'OA', | ' Ole', | ' Ale', | ' TL', | 'APE', | ' Tol', | ' Salvador', | 'Ale', | ' Sul', | ' SAL' |
| 'BT', | ' Damascus', | 'PID', | 'ISP', | 'BD', | ' Wis', | 'IBLE', | 'BI', | ' Brady', | ' Illinois' |
| 'India', | ' India', | ' IA', | ' EL', | ' Modi', | ' JE', | ' Indians', | ' AE', | ' Io', | ' Wick' |
| ' Os', | 'OSS', | 'ei', | 'ees', | ' Pirates', | 'EE', | 'e', | 'Os', | 'OS', | 'OG' |
| 'vb', | ' Bor', | 'VB', | 'pnt', | ' BA', | ' Bil', | 'BER', | ' Bir', | 'Bir', | ' Brun' |
| 'yg', | 'yy', | 'YS', | 'YP', | 'yc', | 'isi', | 'wei', | ' Feld', | 'Eh', | 'Y' |
| 'PY', | ' Zam', | ' Tay', | 'Ay', | ' Ky', | ' y', | ' Ay', | 'Ky', | ' Theo', | 'Py' |
| 'nj', | ' Egg', | ' Jets', | ' eggs', | ' Yuan', | 'ECD', | 'Io', | ' egg', | ' IJ', | ' ERR' |
| 'FER', | ' Fran', | 'rf', | 'Fran', | 'IF', | ' iT', | ' Fang', | 'fi', | 'RF', | 'RL' |
| 'AW', | ' AW', | 'AQ', | 'QA', | 'aq', | 'alias', | ' AA', | ' AO', | ' BA', | ' Falk' |
| 'ticos', | 'xB', | 'TES', | ' IB', | 'ross', | 'IRST', | 'chos', | 'abis', | 'oracle', | 'robl' |
| ' Hogan', | 'INGTON', | ' Hyde', | ' GT', | 'anson', | ' Duncan', | 'ISPR', | ' Roland', | ' GD', | ' Dum' |
| 'ember', | ' Indians', | 'aste', | ' Gem', | 'Ind', | 'stem', | 'ruby', | ' Ghana', | 'estead', | ' Rails' |
| 'ORK', | 'agma', | 'pb', | 'iq', | 'mr', | 'illo', | 'MSO', | 'ork', | 'mq', | ' Amend' |
| ' Rom', | ' BB', | ' rom', | 'Rom', | 'BR', | ' Rams', | ' Ramos', | ' BR', | ' Rangers', | 'BB' |
| 'tics', | 'iast', | ' n\u00FA', | 'gue', | ' MN', | 'inx', | 'ilor', | ' concess', | 'yc', | 'CTOR' |

Table 3: The top 10 decoded tokens for each right singular vector from the memory head 15.7

| | | | | | | | | | |
|---|---|---|---|---|---|---|---|---|---|
| ',', '.;', '.\u200B', '.*', '.:', '.-', '.\)', '.?', '.);', '.).' | | | | | | | | | |
| 'ilogy', 'vex', '\u5FC5', 'xspace', 'verages', 'loat', '\uFFFD', '\u3083', ' cres', 'HPP' | | | | | | | | | |
| 'tron', '.%', '.\_', '———', ' Salem', ' Telesc', 'bsy', '','', 'olean', 'inn' | | | | | | | | | |
| 'ometown', 'LLY', 'suit', '00000000', ' Caption', ' lib', 'ETHERTYPE', 'velt', 'ESULT', 'oxic' | | | | | | | | | |
| '.', '..', '..', ', 'hers', ' DSL', 'GHz', ' VALUES', '..'', 'mic', ' Experiment' | | | | | | | | | |
| ' Integr', 'Ob', ' Lands', ' harass', 'whe', 'ess', ' Land', 'obenz', 'acks', ' lord' | | | | | | | | | |
| 'hea', 'omics', 'olu', 'xa', 'imus', 'Ui', 'irement', 'ilder', 'uren', 'coin' | | | | | | | | | |
| 'lor', ' Shot', 'icult', '...', 'java', 'CUIT', 'iento', ' Secondary', 'Secondary', 'iday' | | | | | | | | | |
| 'ITED', 'odel', 'oda', 'bench', 'bie', ' peninsula', 'olan', 'igr', 'pres', 'itable' | | | | | | | | | |
| 'ano', 'boro', 'Tg', 'TN', 'prises', 'bil', 'gen', ' ! ! !', 'heimer', ' Gen' | | | | | | | | | |
| ' ups', 'skins', 'dead', 'ranging', 'slant', ' JD', 'posts', 'PUBL', 'thood', 'arshal' | | | | | | | | | |
| 'quo', ' Sign', 'ibi', 'express', 'sign', 'corner', 'itten', 'furt', 'alam', '\u53F7' | | | | | | | | | |
| ' Karn', ' chains', ' delays', ' Sout', ' 549', ' delay', 'hog', 'JavaScript', 'NAME', 'ember' | | | | | | | | | |
| 'Transport', 'Tra', 'bus', ' Luc', 'L', ' L', 'cr', ' Del', ' Gl', 'ask' | | | | | | | | | |
| ' .', '(.', ' GB', '\u0101r', 'GB', 'opan', 'azol', 'r\u00E4', 'SUM', '.&' | | | | | | | | | |
| 'rier', 'extensions', 'jh', 'AUD', 'oda', ' scler', 'PLC', 'pie', 'ROW', 'root' | | | | | | | | | |
| 'omin', '\uFFFD \uFFFD', 'osin', 'ys', ' Reuters', 'yzed', 'DLL', 'ctive', 'GV', ' content' | | | | | | | | | |
| 'rient', 'olev', 'gen', 'urd', 'LAY', 'IENT', 'inus', 'heed', ' al', 'ZH' | | | | | | | | | |
| 'rial', ' precision', 'ret', 'feet', 'st', ' WM', 'Execution', 'MC', '\u00E2t', 'oc' | | | | | | | | | |
| ' Mars', 'half', ' half', 'in', ' Notes', ' privately', 'URN', ' halves', 'Execution', ' Spar' | | | | | | | | | |
| 'esan', 'udson', 'rese', 'nar', 'CHANT', ' hooked', 'onium', 'gus', 'Orientation', 'esium' | | | | | | | | | |
| 'azer', 'cons', 'orno', ' deput', ')\u2013', 'gtr', 'uffix', 'iele', 'ennas', 'din' | | | | | | | | | |
| 'Kay', 'FT', 'itor', 'oda', 'izards', 'xym', 'raj', ' aster', 'IRST', 'gether' | | | | | | | | | |
| '¡¿', '¿::', 'erton', 'cats', 'spec', 'bo', 'cA', 'hk', 'h\u00E4', ' curv' | | | | | | | | | |
| 'teenth', 'deal', 'aughters', 'xsl', ' check', 'abin', 'datab', ' Furn', 'ots', 'ride' | | | | | | | | | |
| 'Cur', ' sch', 'A', 'Gh', 'inetics', ' possession', ' knob', ' thousands', 'aler', ' favour' | | | | | | | | | |
| '\u671F', ' DEAL', 'otto', 'uff', ' decks', 'nolimits', 'assign', 'afen', 'deal', ' reload' | | | | | | | | | |
| 'shots', 'CRA', 'rim', 'ARS', ' bonds', ' $@', ' Buch', ' incumbent', ' contacts', 'ars' | | | | | | | | | |
| '-¿', 'ango', 'eti', ' Procedure', 'k\u00E4', 'beh', '-¿\_', ', ']-¿', ' Entry' | | | | | | | | | |
| 'J', 'uppose', ' G', ' Coin', ' risk', '[[', 'mathb', 'coin', ' Jac', ' trail' | | | | | | | | | |
| ' K', ' k', 'kubernetes', 'ocent', ' Mats', 'rels', 'aren', 'nger', 'intf', 'izione' | | | | | | | | | |
| 'POSE', 'obox', '7554', 'ivers', 'ai', 'eral', 'Ax', 'AUX', 'ahi', 'lett' | | | | | | | | | |
| 'VERTIS', 'pendicular', 'OF', 'new', 'zel', 'hores', 'aser', 'fills', ' establ', 'del' | | | | | | | | | |
| ' Rate', 'unks', 'rors', 'RATE', 'ITC', 'acional', 'IBLE', 'Rate', ' Jen', 'TL' | | | | | | | | | |
| 'api', 'Vill', 'pec', 'inction', 'iere', 'raw', ' Wis', ' CTL', 'ources', ' determin' | | | | | | | | | |
| ' vacated', ' vacate', 'cos', ' competitor', 'rtl', 'ather', 'floor', 'Cos', 'below', '@' | | | | | | | | | |
| 'ende', 'ilor', 'velle', ' Licensed', ' ai', 'ai', ' transition', 'transition', ' surname', ' Wiley' | | | | | | | | | |
| 'leans', 'zing', 'eman', 'agine', 'anca', 'adow', 'ahan', ' Vu', 'elta', 'anic' | | | | | | | | | |
| ' Eug', '\uFFFD', ' Thames', 'Resolver', 'ulic', 'chro', '\u00F4', 'gels', 'oku', 'EMENT' | | | | | | | | | |
| 'idden', 'ugs', 'Pix', ' pla', ' lid', 'untu', 'upe', 'unds', 'Uri', 'omi' | | | | | | | | | |
| 'rite', 'hem', 'RR', 'ONS', 'hib', ' Hos', '\u307E \u305B', 'nice', ' INS', '\uFFFD' | | | | | | | | | |
| '@', 'ERTYPE', 'minimum', '\u308D', 'ugu', ' Coy', '\u00F8re', 'unde', 'fff', ' Buffalo' | | | | | | | | | |
| ' monot', 't', ' pace', 'ione', ' denied', 'ening', 'ST', ' Rot', 'ahan', 'cons' | | | | | | | | | |
| 'front', 'ries', 'encial', 'artifactId', ' Reader', ' Set', 'osis', 'container', 'llo', 'rg' | | | | | | | | | |

| |
|---|
| ' Meet', 'Category', 'seek', 'MIX', 'ambers', 'Meet', 'category', ' Articles', 'onitrile', 'umns' |
| 'qrt', 'album', 'ipart', 'ERE', ' MAC', 'msgstr', 'Winter', 'ENS', 'letal', 'eville' |
| 'enda', ' Za', 'ipro', 'NOS', ' Va', ' ectopic', ' Accept', 'decor', '''
}](#', '\uFFFD' |
| '\u043B \u044E', 'opyright', 'nex', 'cept', '\u043E \u0431 \u044B', 'itel',
'olecule', 'YY', 'azol', 'omic' |
| 'yset', 'isan', 'APS', ' Fly', 'REC', 'apolis', ' Pont', 'anos', 'hov', 'Face' |
| 'DEV', 'owa', '...
', 'uy', 'swe', ' fd', 'micromachines', 'dev', ' NG', 'rud' |
| 'k\u00E9', 'gle', 'ieri', 'inki', 'attach', ' requ', 'state', 'mak', 'Text', 'State' |
| 'XT', 'roc', ' ==========================================
===================',
'letal', 'rican', 'xt', ' =&', 'elled', 'rapper', ' Sci' |
| 'aptop', 'com', '\uB2C8 \uB2E4', 'npmjs', 'target', 'commit', ' Ct', ' lust', '\u017Ce', 'compact' |
| 'uin', 'printStackTrace', 'ozo', ' Jay', 'uously', 'ford', 'decode', 'ondon', 'iance', 'nbsp' |
| 'ilon', 'yard', 'WARD', 'ovember', ' moonlight', 'yc', 'rud', ' seller', 'IPT', ' immature' |
| ' beyond', ' Rivers', 'your', '[_', 'beyond', ' Tut', 'prob', '#:', ' Claus', ' Gore' |
| '\uFFFD', ' transitions', '\uFFFD', 'orf', 'ECT', ' Mu', 'isu', '¡', 'eng', 'embedded' |
| 'ieg', 'ally', 'IE', 'icus', 'vphantom', 'OC', 'ais', '.").', 'ilus', 'amin' |
| 'CRO', ' ly', '\u00A0 \u00A0 \u00A0 \u00A0 \u00A0
\u00A0 \u00A0 \u00A0 \u00A0 \u00A0 \u00A0
\u00A0 \u00A0 \u00A0 \u00A0 \u00A0 \u00A0
\u00A0 \u00A0 \u00A0 \u00A0 \u00A0 \u00A0
\u00A0 \u00A0 \u00A0 \u00A0 \u00A0 \u00A0 \u00A0 \u00A0 \u00A0',
'eston', 'INVAL', 'fr', ' bead', 'aden', 'ICK', 'slant' |
| '-¿', ' pitch', 'arent', 'pitch', 'pher', ' pitches', '['''', 'iop', 'kowski', 'Vers' |
| 'ucid', 'tty', 'uve', '\u0163', 'Testing', 'getText', ' Carey', 'mys', 'RESULTS', 'precision' |
| ' cure', ' curative', '\u00E4 \u00E4n', 'mathchoice', 'ARR', 'aires', ' super', 'Super', 'Bits', ' sex' |
| 'NAM', 'PR', ' batt', 'asti', 'Names', ' DPP', ' pd', 'NAP', ' illustrated', 'NAME' |
| 'cho', 'code', 'gang', 'chal', 'TF', 'activ', 'rip', '\uFFFD', '\u00F1o', 'och' |

Table 4: The top 10 decoded tokens for each right singular vector from the context head 19.4

,