# OpenReview forum: "Characterizing Mechanisms for Factual Recall in Language Models"
_EMNLP/2023/Conference — EMNLP 2023 Main_

### Official Review · Reviewer_ecsM · 2023-08-04

**Typos Grammar Style And Presentation Improvements:** 235
**Soundness:** 4

**Excitement:**

4: Strong: This paper deepens the understanding of some phenomenon or lowers the barriers to an existing research direction.

**Paper Topic And Main Contributions:**

This paper investigates LLM behavior when facts are injected through prompts to alter model output. This allows updating outdated information and improving factual consistency. This is investigated by having LLMs name capitals of countries with counter factional prompts and quantifying when they draw information from memory or (in)context. Experiments consider two LLMs with different sizes (parameters); Pythia and GPT2 models. The results show that a models tendency to draw from memory is correlated with term frequencies in pre-training data and model size.

The paper proceeds to investigate if it is possible to detect when LLMs draw from memory or context using methods from mechanistic analysis. Results show that model output can consistently be altered to draw from context over memory by identifying and changing a single attention head.

**Questions For The Authors:**

In section 6 you only conduct experiments on large models, which I assume is motivated by the previous sections finding that large systems draw more on memory. However, how does the method described in section 6 influence smaller models? Does it have my effect?

**Reasons To Accept:**

I really enjoyed this paper. This paper highlights a known issue with LLMs and designs well thought out experiments that quantify the phenomenon and proposes a promising solution/research direction. The findings are a much welcomed contribution to LLM research literature and will likely pave the way for future initiatives on the matter. Experiments are well motivated, sound, and the results are convincing.

**Reasons To Reject:**

Although emphasized in the paper, the findings related to altering model behavior (1/2 of the papers contribution) are based on a single simple dataset on the topic of predicting capital city names. Again described in the paper, this alteration did not generalize to a different dataset, leaving questions about why this is the case unanswered. It is unclear from the papers current state whether there is any reason we would think this would generalize to other types of generations. Despite this, the quantification of the in-context/memory phenomenon and showing that it is possible (at least on a one dataset) is incredibly interesting and enable future research to elaborate on the matter.

**Reproducibility:**

4: Could mostly reproduce the results, but there may be some variation because of sample variance or minor variations in their interpretation of the protocol or method.

**Reviewer Confidence:**

4: Quite sure. I tried to check the important points carefully. It's unlikely, though conceivable, that I missed something that should affect my ratings.

---

> ### Author Rebuttal · Authors · 2023-08-29
>
> Thank you for your thorough reviews of our paper. We are very glad you enjoyed our paper. We would like to address some of your concerns.
>
> 1. “the findings related to altering model behavior (1/2 of the papers contribution) are based on a single simple dataset on the topic of predicting capital city names.”
>
> We understand the concern of the lack of generalizability of our selected heads and scale to different dataset. However, we suspect this is likely an issue with the label domain. In our main dataset world capital, the label space is relatively confined to the space of city names. However, in the counterfacts dataset, the label space is very broad (e.g., questions about famous people’s place of birth, geography, etc.) . Previously, Geva, et al 2023 showed evidence that specific attention heads retrieve a certain mapping between a given attribute and a label field. Therefore, for a dataset with labels from multiple domains, it is less likely for one head to be sufficient in retrieving memory.
>
> 2. In section 6 you only conduct experiments on large models, which I assume is motivated by the previous sections finding that large systems draw more on memory. However, how does the method described in section 6 influence smaller models? Does it have my effect?
>
> The reason we chose to focus on bigger models is that we increase the number of “in-context” answer predictions. Increasing the number of predictions towards the “memorized answers” are more difficult since we are uncertain if such knowledge is acquired in pretraining. As seen in Figure 3, for the models of smaller size, a dominating proportion of predictions are already the in-context answers. Therefore, we won’t be expecting to see a huge increase after intervention. For slightly larger models, we observe a decreasing number of in-context answer predictions, therefore we can expect the intervention to play a larger effect.
>
> Thank you again for your reviews and we appreciate the feedback a lot!

---

### Official Review · Reviewer_jXDd · 2023-08-05

**Soundness:** 4

**Excitement:**

4: Strong: This paper deepens the understanding of some phenomenon or lowers the barriers to an existing research direction.

**Missing References:**

these papers might be potentially relevant:

Understanding Transformer Memorization Recall Through Idioms (Haviv et al., EACL 2023)

Counterfactual Story Reasoning and Generation (Qin et al., EMNLP-IJCNLP 2019)

Counterfactual reasoning: Testing language models’ understanding of hypothetical scenarios (Li et al., ACL 2023)

**Paper Topic And Main Contributions:**

The paper investigates how language models resolve the conflict between memorized information consistent with world knowledge and in-context counterfactual information. It provides a dataset consisting of knowledge about world capitals and conducts two sets of experiments. In the first experiment, it uses zero-shot learning to probe the percentage in which factual/counterfactual capitals are generated. LMs show a strong preference toward memorized factual capitals, and this preference is modulated by word frequency. In the second experiment, it uses head attribution to identify memory attention heads and in-context heads. The authors successfully identified memory head and showed scaling a single head would promote in-context answer to 86%. The paper contributes to growing evidence on LMs' conflict resolution between world knowledge and contextual information, and to identifying specialized attention heads.

**Reasons To Accept:**

1. The paper presents a valuable investigation on LMs' behavior when in-context information is inconsistent with stored world knowledge. It adds to the growing body of literature, showing that LMs heavily rely on memorized information to generate predictions. The design is simple and effective.

2. The authors used a simple way to find attention heads that encode memorized and in-context information, respectively. The results on the world capitals dataset show that scaling memorized and in-context heads affect model performance asymmetrically, and that scaling down memory heads could effectively boost model performance. This sheds light on tuning a small amount of parameters to control model behaviours.

**Reasons To Reject:**

1. The observation "as the frequency of the country increases, models are more inclined to predict the memorized answers" seems to be contradictory to the observation in Li, Yu & Ettinger (2023, ACL), where the authors found more robust encoding of world knowledge (cats love meat) would encourage successful counterfactual predictions (if cats were vegetarians, they would love vegetables). I wonder if the strong memorization preference is because the world capital dataset presents a violation to the world knowledge without any verbal counterfactual signals, so the right behaviour is to ignore this 'wrong' information.

2. Though the authors found scaling the memorization head would boost the performance on world capital dataset, this is not generalizable to other counterfactual dataset that shares a very similar structure. In addition, scaling up and down the memory heads show asymmetrical effect on the model performance. These might suggest that the attention heads might encode some dataset-specific strategies, rather than a more broad strategy to balance out in-context and memorized information. For example, the 'in-context head' might encode the tendency to repeat the last word in the previous sentence, rather than to pay more attention to previous context in general.

**Reproducibility:**

4: Could mostly reproduce the results, but there may be some variation because of sample variance or minor variations in their interpretation of the protocol or method.

**Reviewer Confidence:**

3: Pretty sure, but there's a chance I missed something. Although I have a good feel for this area in general, I did not carefully check the paper's details, e.g., the math, experimental design, or novelty.

---

> ### Author Rebuttal · Authors · 2023-08-29
>
> Thank you for your thorough reviews. We find your feedback very helpful for the improvement of our work. Thank you for pointing us to additional references. We would like to address some of your comments.
>
> 1. “The observation "as the frequency of the country increases, models are more inclined to predict the memorized answers" seems to be contradictory to the observation in Li, Yu & Ettinger (2023, ACL)”.
>
> This is a great pointer to related work that we missed. We would like to point out that our results actually align with Li, Yu & Ettinger (2023, ACL) on models with similar sizes. With GPT2, we also found the predictions are heavily biased towards the in context answer (with up to 60%) shown in Figure 3. MPnet, Bert  and Roberta are similar in number of parameters with GPT2. The results differ on the larger Pythia models, which are beyond the scale considered by LYE 2023.
>
> 2.  “this is not generalizable to other counterfactual dataset that shares a very similar structure”
>
> Although the counterfacts dataset has a similar structure, the world capital dataset has a confined and specific label space (capital cities) while the counterfacts dataset has a much larger scope of labels (e.g., questions about places of birth, founders of companies, geography, etc.). As shown by Geva, et al 2023, attention heads typically encode a mapping from an attribute to a specific domain of knowledge rather than a concept like ‘memory’, which our results support.
> For the camera ready, we would like to include an attention head analysis which aims to identify trends in the information encoded in the weights of the heads as well as trends in the attention patterns. This is to address the point of what the attention heads are actually doing. We are finding a trend in which the memory heads we identify encode domain-specific knowledge; in this case, country/city information. We can expand on this further during the discussion if necessary.
>
> 3. “the 'in-context head' might encode the tendency to repeat the last word in the previous sentence, rather than to pay more attention to previous context in general.”
>
> The in-context head is likely encoding a simple ‘copying’ mechanism, however we find it interesting that the simple scaling intervention is enough to predictably control this behavior. We will include results on the copying heads in the previously mentioned head analysis, but we did notice that the intervention sometimes copies just the answer token (“London”) or copies the entire sentence that contains the answer (“The capital of Poland  is London.”), so it is not following one specific pattern.
>
> Thank you again for your review and we appreciate the feedback very much!

---

### Official Review · Reviewer_tNke · 2023-08-11

**Soundness:** 4

**Excitement:**

4: Strong: This paper deepens the understanding of some phenomenon or lowers the barriers to an existing research direction.

**Missing References:**


Hupkes et al., have some works about memorization vs generalization. It is not taken into account "in context" but you might want to be familiar with those works too.

Belinkov and Orgad also have works related to interventions that you might want to include in your related work.

**Paper Topic And Main Contributions:**

Characterizing Mechanisms for Factual Recall in Language Models

The authors investigate the behavior of LMs with respect to context learning vs memorization.
They use a simple dataset (country-capital) and confused the model with conflicting information in the prompting in order to see what the model would prefer - the information in the prompt (in-context) or the world knowledge inside the model (memory).
(the capital of Poland is London. What is the capital of Poland?)

The authors showed that LMs tend to have a preference to use the answer they have memorized. Later they asked if there is a specific mechanism within the model that controls whether the memorized or in-context answer is generated and whether that can be isolated and controlled.

The authors developed a method of intervention and also tried to investigate the robustness of the discovered mechanism, by applying the same scale intervention on a new dataset (which failed and left for future work).

**Questions For The Authors:**

A - How is it that in Figure 2, the values of the in-context answer and the memorized answer do not add up to 1? Is it because there are answers that are neither in-context nor memorized? You did not address this in the paper.

B - The results of the inverse correlation between in-context and in-context capital (or capital and corresponding country) (the two right graphs in Figure 2 is not making sense. You wrote "When the given in-context capital is more prevalent in the training data, for example Beijing, the model tends to predict the memorized answer"  What is the "memorized answer"?  And Why is it happening?

**Reasons To Accept:**


I enjoyed reading this study. This paper has a good and complete story. It is an interesting topic, challenging to attack, and important. The authors succeed to present the topic in a clearer way. The authors made rigorous analyses with many experiments, which test different aspects that give a deep understanding of the problem. The authors exemplify the consequences of this study with "transfer learning" the conclusions and methods to a new task. There were "null results" with using the new method on a new dataset but I do not see it as a problem but as an integrity of research (reporting the results as they are and not trying to sell anything)  and it might be a challenge for future work.


**Reasons To Reject:**

My level of familiarity with the interpretability literature is low, so it is possible that I am missing important references and comparisons with previous works (although I recognized some of the references that the authors suggested and those that I recognized are very relevant and related to the topic).

Also, sections 6.2-6.3 including Figures 4-6 were hard for me to follow - I did not understand exactly everything there. I'm not sure if that is because of the writing or my lack of background.

For the authors - please answer the questions below

**Reproducibility:**

4: Could mostly reproduce the results, but there may be some variation because of sample variance or minor variations in their interpretation of the protocol or method.

**Reviewer Confidence:**

3: Pretty sure, but there's a chance I missed something. Although I have a good feel for this area in general, I did not carefully check the paper's details, e.g., the math, experimental design, or novelty.

**Typos Grammar Style And Presentation Improvements:**

line 203: "the the"
line 227: "We generate the a full sentence"
Figure 2 first line of the caption - "capital 1"?
line 304: "that that"
line 313: "that that"
line 330: "the the"

---

> ### Author Rebuttal · Authors · 2023-08-29
>
> Thank you for your thorough review. We are very appreciative of your feedback, including missing references and pointers to typos. We would like to clarify and address some of your comments.
>
> 1. “Also, sections 6.2-6.3 including Figures 4-6 were hard for me to follow”
>
> We apologize that the figures are hard to follow. In essence, figure 4 shows how we picked which attention head to tune corresponding to section 6.1. Figure 5 shows with the selected attention head, which scale we will tune it with, corresponding to section 6.2. We identify the attention head and scale with a small randomly selected subset of examples. Figure 6 shows how predictions change when we change the selected head output value with the selected scale on all the inputs of the dataset. For example downweighting the memory head leads to fewer
>
> 2. How is it that in Figure 2, the values of the in-context answer and the memorized answer do not add up to 1? Is it because there are answers that are neither in-context nor memorized? You did not address this in the paper.
>
> Yes, this is because some of the answers it produces don’t fall into either category. The following is an example. The predictions is the country name instead of either the memorized answer of the in context answer.
> The capital of Dominica is San José.
> Q: What is the capital of Dominica?
> A: Dominica.
> Q: What is the capital of Dominica?
>
> 3. The results of the inverse correlation between in-context and in-context capital (or capital and corresponding country) (the two right graphs in Figure 2 is not making sense. You wrote "When the given in-context capital is more prevalent in the training data, for example Beijing, the model tends to predict the memorized answer" What is the "memorized answer"? And Why is it happening?
>
> Here, the memorized answer refers to the ground truth capital city of the country; i.e., what the model would have learned in pretraining.
> In the example of:
> “The capital of Poland is Beijing, what is the capital of China?” The memorized answer in this case will be “Warsaw” and the in-context answer is “Beijing”. The point we want to make here is that not only does asking about frequently seen countries (e.g., Poland) make the model more likely to predict the memorized answer (“Warsaw”) in the right two graphs, but using more frequent capital cities as the counterfactual (e.g., The capital of Poland is Beijing [vs. e.g., Bissau]) results in the model being less likely to use the provided in-context answer in the left two graphs. We found this point a little difficult to communicate so we will work on the presentation of this for the camera ready version. Please let us know if this makes more sense now.
>
> Thank you again for your reviews and feedback on our paper!

---

### Meta-Review · Area_Chair_1dv6 · 2023-09-19

**Recommendation:** 5

**Metareview:**

This work explores the effect of knowledge memorized in pretraining by language models vs information that is supplied to them in-context, finding that LMs have a strong preference towards memorized information. The authors also explore interesting questions related to the term frequency of these facts in the pretraining data. All reviewers appreciated this line of investigation, and the rigorous analyses in this work.

---

### Decision · Program_Chairs · 2023-10-07

**Decision:**

Accept-Main

**Comment:**

This work explores the effect of knowledge memorized in pretraining by language models vs information that is supplied to them in-context, finding that LMs have a strong preference towards memorized information. The authors also explore interesting questions related to the term frequency of these facts in the pretraining data. All reviewers appreciated this line of investigation, and the rigorous analyses in this work.